# Subglacial carbonate deposits as a potential proxy for a glacier's former presence

Matej Lipar[1], Andrea Martín-Pérez[2], Jure Tičar[1], Miha Pavšek[1], Matej Gabrovec[1], Mauro Hrvatin[1], Blaž Komac[1], Matija Zorn[1], Nadja Zupan Hajna[3], Jian-Xin Zhao[4], Russell N. Drysdale[5,6], Mateja Ferk[1]

[1]Anton Melik Geographical Institute, Research Centre of the Slovenian Academy of Sciences and Arts, Ljubljana, 1000, Slovenia
[2]Ivan Rakovec Institute of Palaeontology, Research Centre of the Slovenian Academy of Sciences and Arts, Ljubljana, 1000, Slovenia
[3]Karst Research Institute, Research Centre of the Slovenian Academy of Sciences and Arts, Postojna, 6230, Slovenia
[4]School of Earth and Environmental Sciences, The University of Queensland, Brisbane, QLD 4072, Australia
[5]School of Geography, The University of Melbourne, Melbourne, VIC 3053, Australia
[6]EDYTEM, UMR CNRS 5204, Université de Savoie-Mont Blanc, 73376 Le Bourget Du Lac-Cedex, France

*Correspondence to*: Matej Lipar (matej.lipar@zrc-sazu.si)

**Abstract.** The retreat of ice shelves and glaciers over the last century provides unequivocal evidence of recent global warming. Glacierets (miniature glaciers) and ice patches are important components of the cryosphere that highlight the global retreat of glaciers, but knowledge of their behaviour prior to the Little Ice Age is lacking. Here, we report the uranium-thorium age of subglacial carbonate deposits from a recently exposed surface previously occupied by the disappearing Triglav Glacier (southeastern European Alps) that may elucidate the glacier's presence throughout the entire Holocene. The ages suggest their possible preservation since the Last Glacial Maximum and Younger Dryas. These thin deposits, formed by regelation, are easily eroded if exposed during previous Holocene climatic optima. The age data indicate the glacier's present unprecedented level of retreat since the Last Glacial Maximum, and the potential of subglacial carbonates as additional proxies to highlight the extraordinary nature of the current global climatic changes.

## 1 Introduction

Glaciers respond to climatic changes making them valuable archives with which to study the effects of past and current climatic changes (Benn and Evans, 2010). Their worldwide retreat over the last century provides evidence in support of current global climate change even though the decrease of summer insolation in the Northern Hemisphere favours climate cooling (Solomina et al., 2015; IPCC, 2018). The uniqueness of this trend can only be understood when compared to past retreats and advances of different types of glaciers throughout the entire Holocene.

The data that informs a glacier's dynamics in the past can be provided by subglacial carbonate deposits (Hallet, 1976; Sharp et al., 1990). These are thin carbonate crusts formed between the ice and bedrock by regelation at the glacier base on the lee

side of a bedrock protuberance (Hallet, 1976; Lemmens et al., 1982; Souchez and Lemmens, 1985), and can provide information on chemical and physical processes present at the time of their formation (Hallet, 1976).

Unlike other glacial deposits (e.g., moraines), subglacial carbonates may be eroded in a matter of decades (Ford et al., 1970),
suggesting their exposure can be recent. They have been reported from deglaciated areas of northern North America (Ford et al., 1970; Hallet, 1976; Refsnider et al., 2012), northern Europe and the European Alps (Lemmens et al., 1982; Souchez and Lemmens, 1985; Sharp et al., 1990; Lacelle, 2007; Gabrovec et al., 2014; Colucci, 2016; Thomazo et al., 2017), Tibet (Risheng et al., 2003), New Guinea (Peterson and Moresby, 1979) and Antarctica (Aharon, 1988; Frisia et al., 2017). The uranium-thorium (U-Th) method remains the main dating technique for these carbonates, whilst the [14]C technique is invalidated by
modern carbon contamination (Aharon, 1988) and possible dead carbon effect.

The deposits recently exposed by the rapid retreat of the Triglav Glacier (Fig. 1) offer a unique opportunity to gain additional knowledge of this glacier's behaviour in the past. The Triglav Glacier extends from the northeastern side of Mount Triglav (2864 m a.s.l.) in the Julian Alps (Fig. 2, Supp. Fig. S2), Slovenia's highest mountain and consisting of Upper Triassic limestone and dolostone (Ramovš, 2000; Pleničar et al., 2009). This region's montane climate is characterised by precipitation
from moisture-bearing air masses from Mediterranean cyclones, typically in autumn and late spring, and by frequent freeze-thaw cycles (Komac et al., 2020).

At present, the Triglav Glacier is one of only two remaining ice masses in Slovenia since the last extensive Pleistocene glaciation (Bavec and Verbič, 2011; Ferk et al., 2017; Triglav-Čekada et al., 2020). The retreat of the glacier has exposed a glaciokarst environment (Fig. 2, Supp. Fig. S2) comprising a range of erosional (shafts (i.e., vertical caves), karrens, polished
surfaces and roches moutonnées) and depositional (moraines, boulders, till, carbonate deposits) features (Colucci, 2016; Tičar et al., 2018; Tóth and Veress, 2019). The known extent and behaviour of the Triglav Glacier spans from the present to the Little Ice Age (LIA) (Colucci, 2016; Colucci and Žebre, 2016), the cool-climate anomaly between the Late Middle Ages and the mid-19[th] century (Grove, 2004; Nussbaumer et al., 2011), and is based on geomorphological remnants, historical records and systematic monitoring since 1946 (Gabrovec et al., 2014). Over the last 100 years, the glacier retreated from ca. 46 ha
(extending between 2280 and 2600 m.a.s.l.) to ca. 0.5 ha (between 2439 and 2501 m.a.s.l.) (Supp. Fig. S1), with a downwasting rate of around 0.6 m/yr (1952-2016) (Triglav-Čekada and Zorn, 2020). The glacier has thus evolved from the plateau-type through glacieret to the present ice-patch type.

Glacierets are defined as a type of miniature (typically less than 0.25 km$^2$) glacier or ice mass of any shape persisting for at least two consecutive years (Cogley et al., 2011; Kumar, 2011). Serrano et al. (2011) discriminate them from ice patches,
arguing that glacierets are "the product of larger ancient glaciers, still showing motion or ice deformation, although both very low. They have a glacial origin, glacial ice and are never generated by new snow accumulation"; ice patches on the other hand are "ice bodies without movement by flow or internal motion". Despite their small size, glacierets occupy a significant ice volume fraction at regional scales (Bahr and Radić, 2012), and can therefore be considered as an important target for palaeoclimate studies. Accordingly, many present-day glacierets are closely monitored and studied (Gądek and Kotyrba, 2003;
Grunewald and Scheithauer, 2010; Gabrovec et al., 2014; Colucci and Žebre, 2016), but the peculiarity of current global

climate change requires more evidence from different proxies and from the past when current ice patches and glacierets were still glaciers.

Here, we present preliminary geochemical and petrological data of subglacial carbonate deposits recently exposed by the retreat of the Triglav Glacier. The aim is to highlight the occurrence of deposits in terms of their possible preservation since the Last Glacial Maximum (LGM), discuss the complexity of the deposit and validate the results in the context of the present climate regime of rising temperatures and global retreat of glaciers.

## 2 Methods

The extent of the Triglav Glacier has been measured annually since 1946 and systematically photographed since 1976 (Meze, 1955; Verbič and Gabrovec, 2002; Triglav Čekada and Gabrovec, 2008; Triglav-Čekada et al., 2011; Triglav-Čekada and Gabrovec, 2013; Gabrovec et al., 2014; Del Gobbo et al., 2016), using a panoramic non-metric Horizont camera. The photos were transformed from a panoramic to a central projection in order to allow the calculation of the area and estimation of the volume (Triglav-Čekada et al., 2011; Triglav-Čekada and Gabrovec, 2013; Triglav-Čekada and Zorn, 2020). The early measurement technique was by measuring tape and compass, which enabled measurement of the glacier's retreat from coloured marks on the rocks around the glacier (Meze, 1955). Accurate and continuous geodetic measurements began in the 1990s: standard geodesic tachymetric measurements, UAV photogrammetric measurements (from both the ground and air), GPS measurements, and LIDAR (Triglav Čekada and Gabrovec, 2008; Gabrovec et al., 2014). In addition, several extensive field campaigns were conducted throughout 2018 with the focus on the central part of Triglav Plateau at the side of the present and former glacier (Fig. 1, 2). During this time, five subglacial carbonate samples (Fig. 3) were collected at multiple localities (Fig. 1, 2; Supp. Fig. S2) for laboratory analyses. These were collected 50 m to 100 m from the current ice patch. A hand drill was used to obtain powdered samples for geochemical analysis; sampling was restricted to areas consisting of the densest and thickest sections. Prior to and after each drilling, the surface was cleaned with deionized water and dried under clean air. Drilling was performed with a 0.6 mm drill bit in a clean environment. A stainless steel spatula and weighing paper were used to harvest the drilled powder and transfer it into clean 0.5 ml sterile vials.

Five thin sections (30 - 50 μm) were examined using an Olympus BX51 polarising microscope equipped with an Olympus SC-50 digital camera. Due to the fragility of the samples, they were embedded in Epoxy resin under vacuum before being cut and polished.

The mineralogy of twelve (sub) samples was determined by X-ray diffraction (XRD) using a Bruker D2PHASER diffractometer equipped with an energy dispersive LYNXEYE XE-T detector, located at the ZRC SAZU Karst Research Institute, Slovenia. Powdered samples were scanned from 5 to 70° 2θ at a 0.02° 2θ/0.57 s scan speed. The diffractograms were interpreted using EVA software by Bruker (DIFFRACPlus 2006 version).

Five (sub) samples were dated by the U-Th method at the University of Queensland, Australia. To ensure that samples were suitable for U/Th analysis, they were first measured by ICP-MS for their trace element concentrations. U-Th age dating was carried out using a Nu Plasma multi-collector inductively-coupled plasma mass spectrometer (MC-ICP-MS) in the Radiogenic Isotope Facility (RIF) at the School of Earth and Environmental Sciences, The University of Queensland. Ages were corrected for non-radiogenic $^{230}$Th incorporated at the time of deposition assuming an initial $^{230}$Th/$^{232}$Th ratio of $0.825 \pm 50$ % (the bulk-Earth value, which is the most commonly used for initial/detrital 230Th corrections.. Age errors are reported as $2\sigma$ uncertainties. Full details of the method are provided in the supplementary material - Table S1.

The stable-isotope composition ($\delta^{13}$C and $\delta^{18}$O) of five (sub)samples was measured at the School of Geography, University of Melbourne, Australia. Analyses were performed on $CO_2$ produced by the reaction of the sample with 100% $H_3PO_4$ at 70°C using continuous-flow isotope-ratio mass spectrometry, following the method previously described in Drysdale et al. (2009) and employing an Analytical Precision AP2003 instrument. Results are reported using the standard $\delta$ notation (per mille ‰) relative to the Vienna PeeDee Belemnite (V-PDB scale). The uncertainty based on a working standard of Cararra Marble (NEW1) is 0.05‰ for $\delta^{13}$C and 0.07‰ for $\delta^{18}$O.

## 3 Results

The subglacial carbonate deposits occur on the lee sides of small protuberances on a bare polished and striated limestone bedrock surface near the Triglav Glacier. They are most abundant on the bedrock recently uncovered by the retreating ice, and their occurrence rapidly decreases with the distance from the edge of the present glacieret margin. The fluted and furrowed crust-like deposits are brownish, greyish or yellowish in colour. The deposits do not exceed 0.5 cm in thickness and are occasionally internally laminated.

### 3.1 Mineralogical and petrographic data

X-ray diffraction (XRD) analysis shows that the carbonate deposits mostly consist of calcite, and mixtures of calcite with small amounts of aragonite. XRD also confirms the calcite composition of the host rock (Supp. Fig. S5). The petrographic study has allowed the identification of different fabrics (Fig. 4). Due to the similarities of subglacial carbonate textures to those of speleothem deposits, we have used, where possible, the formal terminology of Frisia and Borsato (2010):

*Primary calcite fabrics* are composed of transparent crystals with uniform extinction. The first crystals to form directly over the bedrock are short columnar (length to width ratios < 6:1) crystals from 50 to 200 µm long (Supp. Fig. S6 and S7). Columnar (L/W ratios ~ 6:1) and elongated columnar (L/W ratios > 6:1) crystals up to 2 mm long and 0.5 mm wide constitute the most abundant fabric. Calcite crystals grow perpendicularly to the substrate on the steeper, nearly vertical areas of the bedrock (Fig. 4a) while in the less steep, nearly horizontal areas, crystals grow inclined, oriented downslope, presumably in the sliding direction of the forming glacier (Supp. Fig. S6). In some areas, the younger crystals crosscut the main direction of the crystal growth of the previous layer, resulting in crystal boundaries resembling unconformities (Supp. Fig. S6).

*Primary aragonite fabrics* consist of acicular crystals (L/W ratio >>6:1) generally growing outwards from a common point in the shape of fans. In some areas of the crusts, these fans are aligned in bands interlayered with very dark, dense micrite and transparent equidimensional calcite crystals, forming layered textures (Fig. 4b, Supp. Fig. S6d).

*Aragonite-to-calcite replacement fabrics* are characterised by calcite crystals of variable size and patchy extinction patterns. They contain abundant fibrous inclusions interpreted as aragonite relicts that are either aligned in layers or unevenly distributed throughout, which results in very irregular textures (Supp. Fig. S6d, e and f). Micrite and microsparite are often associated with aragonite relicts.

## 3.2 Geochemical data and ages

Stable carbon and oxygen isotope ratios ($\delta^{13}C$ and $\delta^{18}O$) of the subglacial carbonate yielded average values of 1.35 ± 0.05 ‰ for $\delta^{13}C$ and -4.32 ± 0.07 ‰ for $\delta^{18}O$ (relative to the V-PDB; Fig. 3).

Whilst the expected ages of subglacial carbonates were of the LIA, the U-Th ages yielded considerably older ages: 23.62 ka ± 0.78 ka, 18.45 ka ± 0.70 ka and 12.72 ka ± 0.28 ka, respectively (Supp. Table S1). The results indicate that these subglacial carbonate dates fall within the LGM and the Younger Dryas (YD). In addition, two U-Th ages corresponding to samples drilled in more surficial calcite layers of the thickest obtained sample yielded 3.85 ka ± 0.09 ka and 1.96 ka ± 0.04 ka, indicating the presence of a glacier of sufficient thickness to cause regelation also during these periods. For details where the samples were obtained from, the reader is referred to Figure 3.

## 4 Subglacial carbonate deposits

The fluted and furrowed morphology of carbonate deposition parallel to the apparent former ice-flow direction, some of the crystal textures, the location of the samples and the age data imply a subglacial origin of the carbonate crusts. The presence of ice-flow oriented calcite crystals suggests that precipitation was strongly influenced by the mechanical force of the ice movement. Similar fabrics occur in different types of tectonic veins, for instance, slickensides, where crystals grow obliquely to the sheets in shear veins, indicating the general shear direction (Bons et al., 2012). Previous studies on subglacial carbonates have considered the orientation of calcite parallel to the ice flow as evidence of its growth in a thin water film confined by sliding regelation ice (Hallet, 1976), and deformation structures such as folds and fractures, the result of glacially imposed stress (Sharp et al., 1990).

The variability of fabrics of the studied crusts and, especially, the presence of aragonite coexisting with calcite indicate spatial and/or temporal variability in local subglacial water chemistry. The most important parameters controlling the precipitation of aragonite versus calcite appears to be the $CaCO_3$ saturation state, the Mg/Ca ratio in the waters (De Choudens-Sánchez and González, 2009) and/or rapid $CO_2$ degassing rates (Fernández-Díaz et al., 1996; Jones, 2017). In freshwater systems like spring deposits (Jones, 2017), and specially in speleothems, Mg/Ca ratios seem to be the main factor controlling aragonite vs calcite precipitation (Frisia et al., 2002; Wassenburg et al., 2012; Rossi and Lozano, 2016). The studied deposits grow over carbonate

bedrock with prevailing limestone and some dolostone (Jurovšek, 1987; Ramovš, 2000), so high Mg/Ca ratios in the water may be partially responsible for the precipitation of aragonite. Similarly, precipitation of aragonite in subglacial deposits at Vestfold Hills, Antarctica, seems to have been favoured by the presence of $Mg^{2+}$ in the waters, mobilised from the pyroxenes of the gneiss bedrock (Aharon, 1988).

The U-Th ages of LGM and YD are in good accordance with the glacier's history when it was expected to be thick enough to cause regelation. However, aragonite-to-calcite replacement fabrics raise the difficult question of whether all subglacial carbonate crystals were primarily aragonite and were subjected to complete recrystallisation to calcite without leaving traces of replacement fabrics, which may provide a false identification of calcite fabrics as primary. This, in turn, may lead to inaccurate interpretation of the U-Th ages (Bajo et al., 2016). Previous studies on corals and speleothems have shown that the diagenetic transformation of aragonite into calcite can affect the accuracy of U-Th dating (Ortega et al., 2005; Scholz and Hoffmann, 2008; Lachniet et al., 2012; Bajo et al., 2016; Martín-García et al., 2019) due to uranium loss, which usually leads to older-than-true U-Th ages. However, the extent of uncertainty can be highly variable depending on several factors such as the initial amount of aragonite and the timing of its transformation to calcite (Bajo et al., 2016), the possible additional redistribution of Th (Ortega et al., 2005; Lachniet et al., 2012) and the extent of chemical exchange with younger subglacial meltwaters (Ortega et al., 2005; Martín-García et al., 2019).

Notably, the U concentration (in ppm; Supp. Table S1) in the youngest sample (2ka; T.03_b1) within this study is 1.77 ppm, whereas it is around 0.41 and 0.46 ppm in two of the old samples, T.01_a1 and T.03_a1, respectively LGM and YD. This could indicate U-loss during aragonite-to-calcite recrystallisation and consequently provide older-than-true ages, or to the contrary, that these two samples represent primary calcite with non-preferential incorporation of U in calcite with respect to aragonite and thus proving to be the most reliable. Assuming the first possibility, the oldest sample (24 ka; T.05) has relatively high U concentration (1.33 ppm), which would indicate less loss of U but still provide data that subglacial carbonates may date back to the last glacial period. Assuming the second possibility, the ages of T.03_a1 and T.03_a2 in correct stratigraphic order would strengthen the reliability of the results. Aharon (1988) used U-Th method for dating two samples with a combination of aragonite and calcite fabrics with high but also variable U content (23.2 and 41.4 ppm), however, the indistinguishable dates compared to the those from pure aragonite led to the conclusion that they provide the reliable age estimates. Nevertheless, due to the external Th incorporated into the samples, he regarded the dates as "maximum ages", which is the appropriate approach also with the dates discussed in this study. Similar U-Th ages (19 - 21 ka) were reported also from northern Canada (Refsnider et al., 2012), however, the carbonate fabrics were not described. The U-Th dates from Antarctica by Frisia et al. (2017) and radiocarbon dates from the French Alps by Thomazo et al. (2017) were performed on calcites. More studies should therefore be done in this direction to positively confirm the LGM and YD ages of carbonates, and especially to gain the high-resolution dates to construct the whole timeline of subglacial carbonate precipitation. Nevertheless, since three of our dates fall in the period of 12 - 24 ka, we will proceed with the discussion of their susceptibility to weathering and its implication to glacier's existence.

The cold Alpine environment during the glacial period with low biological respiration rates could be indicated in the relatively high $\delta^{13}C$ signal, also reported by others (Lemmens et al., 1982; Fairchild and Spiro, 1990; Lyons et al., 2020), but the (re)freezing of the subglacial water causes supersaturation with respect to carbonate and the non-equilibrium conditions produced by this process can affect the stable isotopic composition of the subglacial carbonate, usually leading to isotopic enrichment in the carbonate minerals (Clark and Lauriol, 1992; Courty et al., 1994; Lacelle, 2007). In any case, the climate during the precipitation of subglacial carbonate had to be "warm" enough to produce subglacial water at the glacier-bed. Kuhlemann et al. (2008) suggested a lowering of summer temperature of about 9 - 11°C at the LGM peak, meaning that the summer temperatures at Triglav Glacier should have been around -4.7 to -6.7°C (based on the present summer temperatures at around +6°C (Slovenian Environment Agency, 2020) and considering the recent warming of up to +2.03°C in the area (Colucci and Guglielmin, 2015; Hrvatin and Zorn, in press)), which would have been conditions for the cold base glacier and the absence of subglacial water flow. Whilst small scale detailed palaeoclimate conditions in the southeastern Alps are still uncertain, the current U-Th ages of subglacial carbonates within this research fit with the temporary Garda Glacier (Italy) withdrawal phases reported by Monegato et al. (2017): 23.62 ka age of subglacial carbonate relates to the withdrawal phase between 23.9 and 23.0 ka, and 18.45 ka could relate to the glacier retreat from around 19.7 to 18.6 ka or even the final collapse of the glacier around 17.7 to 17.3 ka. Both time periods could relate to commencement of subglacial water flow also at the Triglav Glacier and consequently the precipitation of subglacial carbonates. In addition, the 12.72 ka age marks the early phase of YD cooling (12.9-11.5 ka ago) (Renssen and Isarin, 1997; Alley, 2000; Broecker et al., 2010).

The $\delta^{18}O_{PDB}$ of subglacial carbonate can be transformed into $\delta^{18}O_{SMOW}$ using the equation $\delta_{SMOW} = 1.03037\ \delta_{PDB} + 30.37$ (Faure, 1977). The mean values of subglacial carbonate $\delta^{18}O_{SMOW}$ would therefore be 25.92‰, ranging from 27.44 to 24.75‰. Using the fractionation factor of 1.0347 for calcite and water at 0°C (Clayton and Jones, 1968) and the $\delta_{SMOW}$ values of subglacial carbonate, we obtain $\delta^{18}O_{SMOW}$ values ranging from -7.02‰ to -9.62‰. On the other hand, if assuming calcite crystals could have all originated primarily as aragonite crystals, we can use the fractionation factor of 1.0349 for aragonite and water at 0°C (Kim et al., 2007) and obtain relatively similar $\delta^{18}O_{SMOW}$ values ranging from -7.21‰ to -9.81‰. This relates to the average Triglav Glacier ice meltwater values ($\delta^{18}O_{SMOW} = -9.3$‰) measured in the summer 2018 (Carey et al., 2020). However, it is slightly heavier when compared to glacier ice samples, which range between -10.0‰ and -12.7‰. The reason could be that water in equilibrium with the growing ice is progressively impoverished in heavy isotopes (Jouzel and Souchez, 1982), or simply that the remnants of residual ice has a different isotopic ratio than the basal ice at the time of carbonate precipitation, which itself could represent a large range in values (Lemmens et al., 1982). In addition, $\delta^{18}O$ differences of a few per mills in the carbonate precipitate can also arise due to variations in subglacial hydrology shifting from closed to open geochemical systems (Hanshaw and Hallet, 1978). However, a geochemical study of the present Triglav ice area (Carey et al., 2020) shows that ice samples within the cave of Triglavsko Brezno Shaft (see Fig. 1 for location) are much lighter in deuterium than those from the Triglav Glacier, which suggests that they were deposited during colder times, indicating they are remnants of older glacier ice.

## 5 Implications for continuously existing glacier during the Holocene

The LGM and YD ages of the Triglav subglacial carbonates provide the first physical evidence that the Triglav Glacier persisted through the Holocene to the present day. Glacierets of southern Europe, including the Triglav Glacier, have generally been viewed as relicts of the LIA, suggesting a discontinuous presence following the Holocene Climatic Optimum (HCO) (Grunewald and Scheithauer, 2010), a period of high insolation and generally warmer climate between 11,000 and 5,000 years ago (Renssen et al., 2009; Solomina et al., 2015).

Being prone to fast weathering (Ford et al., 1970), subglacial carbonate deposits are generally found only on recently deglaciated areas (Ford et al., 1970; Sharpe and Shaw, 1989). The Triglav Glacier area is no exception. The fact that they have not been reported previously in the literature (Meze, 1955; Šifrer, 1963, 1976, 1987) until the year 2005 (Hrvatin et al., 2005; Gabrovec et al., 2014) reinforces their likely recent exposure. Whilst Refsnider et al. (2012) discussed the preservation of the subglacial carbonates by very cold and dry Arctic climate, this is unlikely to be the case of the plateau of the Triglav Glacier, either in the last century or during any possible period of absence of an ice body, because of the large mean annual precipitation at the site (2038 mm; recorded at Kredarica hut, 2515 m a.s.l.) (Slovenian Environment Agency, 2020) (see Fig. 2 for location). This is supported by theoretical numerical data for the chemical denudation of the subglacial carbonate, which is based on denudation rates of limestones (Table 1). Chemical denudation rates on carbonate rocks can vary from ca. 0.009 to 0.14 mm/year (Gabrovšek, 2009). Taking the low and high extreme values for, e.g., 6 ka during the HCO, surface lowering would be between 54 and 840 mm, so the exposed 5 mm thick subglacial carbonate would have been denuded by this time.

In addition, carbonate surfaces in periglacial areas are exposed not only to chemical weathering but also to intensive frost weathering, promoting physical disintegration of minerals (Matsuoka and Murton, 2008). Therefore, had the subglacial carbonate been exposed in the past, it would be expected to be eroded by dissolution or frost weathering. This indicates, that subglacial carbonate was constantly covered with glacier ice, and, as Steinemann et al. (2020) has demonstrated, that glacial abrasion on carbonate bedrock is minimal compared to the abrasion on crystalline bedrock. This would promote the preservation of subglacial carbonates on limestone plateaus in the Alps. Moreover, since the subglacial carbonates form in lee positions of bedrock protuberances, it is likely that abrasion would vanish as abrading rock fragments diverge from the bed at sites of subglacial precipitation due to regelation ice growth.

Organic matter (charcoal/wood) from a non-vegetated, scree-covered moraine ca. 300 m below the main ice patch of the Triglav Glacier (as it stood in the year 2006) was analysed in the radiocarbon ([14]C) laboratory in Erlangen, Germany. The [14]C result yielded 5604-5446 BP age, which provides additional evidence of pre-LIA, and post-LGM and post-YD ice cover (unpublished analysis by Karsten Grunewald and his team). Similarly, the two younger U-Th ages obtained within this study (3.85 ka and 1.96 ka) also provide evidence of a pre-LIA ice cover.

## 6 Glacier variations and palaeoclimatic implications

The complex global pattern of Holocene glacier fluctuations indicates the influence of multiple climatic mechanisms and that individual glaciers may not respond uniformly to a particular set of climate forcings (Kirkbride and Winkler, 2012; Solomina et al., 2015). In addition, glaciers are also influenced by topographic conditions (DeBeer and Sharp, 2017), as well as their size and flow dynamics (Sugden and John, 1976; Nussbaumer et al., 2011). The Alps has experienced several glacial advances and retreats during the Holocene (Nussbaumer et al., 2011), with reports that some glaciers were even smaller than today or absent (Leemann and Niessen, 1994; Hormes et al., 2001; Ivy-Ochs et al., 2009; Solomina et al., 2015). On the other hand, certain regions show evidence of unprecedented modern retreat of glaciers beyond their previous Holocene minima (Koerner and Fisher, 2002; Antoniades et al., 2011; Miller et al., 2013).

Current climate near the Triglav Glacier is characterised by rising temperatures in the ablation season from May to October (Supp. Fig. S3) and a descending trend of the highest seasonal snow elevation (Supp. Fig. S4). If subglacial deposits indicate its ongoing existence throughout the Holocene, the recent retreat of the Triglav Glacier suggests regional climate during previous Holocene optima was cool enough to sustain the glacier, unless an additional (presently unknown) forcing component prevailed which was not important in the past. Exceptionally rapid 21[st]-century melting, for example, has been reported also from Barnes Ice Cap in northern Canada (Gilbert et al., 2017), in the Alps, where the exceptional retreat has been attributed to deposition of industrially sourced black carbon (Painter et al., 2013), and is also evident by a large number of recently melt-out archaeological and paleontological finds (Mol et al., 2001; Basilyan et al., 2011), including the famous "Ice Man" Ötzi in the Tyrolean Alps (Baroni and Orombelli, 1996; Solomina et al., 2015).

Local physiographic influences can insulate small glaciers from the warming effects of regional and global climate (Grunewald and Scheithauer, 2010); it has been shown that some small glaciers are less sensitive to climate fluctuations than previously thought (Colucci and Žebre, 2016). The natural resilience of the Triglav Glacier is due to its relatively high elevation, bright limestone substrate with higher albedo, the vertical water drainage through karstified rocks and the consequent lack of subglacial lakes as heat collectors, and avalanche feeding (Grunewald and Scheithauer, 2010; Gabrovec et al., 2014). Assuming these factors were relatively constant throughout the Holocene, the possible unprecedented retreat may highlight the consequences of direct anthropogenic forcing (Solomina et al., 2015). On the other hand, further comparative research on other small glaciers is needed to test palaeoclimatic data due to the geomorphological peculiarity of glacier regions. For example, despite of the apparent resilience of the Triglav Glacier until the recently, its retreat has been more evident than that of the glacierets/ice patches of Canin (Italy), Montasio West (Italy) and Skuta (Slovenia) (all in the southeastern Alps), which are all lower in elevation than the Triglav Glacier, but with large differences in mean annual precipitation (e.g., in the Triglav area water equivalent precipitation (2038 mm) is 62% of that in Canin area (3335 mm)); potential annual solar radiation also plays a major role in controlling for differences in glaciers' dynamics (Colucci, 2016).

## 7 Conclusion

Three U-Th ages of subglacial carbonate exposed by the retreating Triglav Glacier fall within the LGM and YD, providing the first direct evidence of the Triglav Glacier at that time. The high erodibility of these deposits, once exposed, strongly suggests a continuous glacier cover since their deposition throughout all but the most recent part of the Holocene, including the HCO. This defines the subglacial carbonates as a complex, but an important palaeoenvironmental proxy and a research subject for further analyses with great potential that may further highlight the extraordinary nature of the current global warming in the context of the Holocene.

Several separate lines of additional evidence could strengthen this assertion:

A - Identification and geochemical analyses of primary aragonite would contribute data on primary U concentration in samples (where there is evidence of primary aragonite), which can be used to detect U loss. In situ U-series dating by laser-ablation would represent the best approach as the samples (and laminae) are small and the aim is to target different crystal fabrics, but this is challenging due to low U concentrations.

B - Use $^{36}$Cl nuclide dating method on the limestone hosting the subglacial carbonate deposits to extract the exposure time of the limestone surfaces hosting the carbonate crusts (i.e., no glacier cover) and provide additional data as to whether or not those areas have experienced multiple episodes of exposure during the HCO. The production rate at 2500 m a.s.l. (the mean altitude of the upper part of the present Triglav Glacier) is relatively high, which is an advantage for dating young exposures. In addition, calcite and limestone are one of the best mineral/rock systems for $^{36}$Cl because they often have low $^{35}$Cl abundances, so there is less uncertainty regarding contributions from factors that affect thermal neutron production of $^{36}$Cl, such as water or snow shielding (Fabel and Harbor, 1999; Marrero et al., 2016).

C - The analyses of material on the small remaining ice masses and, if possible, ice cores can highlight the existence of a present-day input to accelerate melting. For example, black carbon and other mineral dust particles in glaciated regions, which accumulate on the ice, can accelerate the melt of glaciers by reducing albedo (Ramanathan and Carmichael, 2008; Ming et al., 2013; Painter et al., 2013; Gabbi et al., 2015).

D - Numerical data on frost weathering. There is a relatively extensive literature concerning the chemical denudation and glacial erosion of the limestone, but scarce numerical data for frost weathering, and no direct measurements concerning subglacial carbonates. This can be experimentally studied by direct freezing-thawing experiments with controlled humidity and additional control for the type of porosity and pressure gradient (Ducman et al., 2011).

**Data availability**

The live and recent-archive photos of Triglav Glacier observation: http://ktl.zrc-sazu.si/

The climate data are available through the Slovenian Environment Agency web page: https://www.arso.gov.si/en/ and http://meteo.arso.gov.si/met/sl/archive/

Additional Triglav Glacier measurement data are available on the dedicated Slovenian Environment Agency web page: http://kazalci.arso.gov.si/en/content/triglav-glacier

All visual computer data are included in Supplementary files.

## Author contributions

ML designed the research, led the study, drafted the manuscript and generated figures. AMP performed petrographic analysis, generated the petrographic figures, contributed the writing and editing of the manuscript. JT, MG, MH, BK, MZ contributed the writing and editing of the manuscript. MP contributed the writing and editing of the manuscript and compiled the
monitoring data. NZH performed XRD analysis. J-XZ performed U-Th analysis and wrote the U-Th methods section of the manuscript. RND performed stable isotope analysis, edited and reviewed the manuscript. MF contributed the writing and editing of the manuscript, overall editing and internal review.

## Competing interests

The authors declare that they have no conflict of interest.

**Acknowledgement**

We thank Kata Cvetko-Barić from the ZRC SAZU Institute of Palaeontology for preparation of the geological thin sections, Manca Volk Bahun from the ZRC SAZU Anton Melik Geographical Institute for producing the map of the sampling areas. We also thank peer-reviewers Renato R. Colucci and Bernard Hallet, and editor Chris R. Stokes for improving the manuscript.

**Financial support**

The work of ML was supported by the European Regional Development Fund: European Union & Republic of Slovenia, Ministry of Education, Science and Sport (2017-2020; research programme OP20.01261) and, including the work of JT, MG, MH, BK, MZ and MP, by the Slovenian Research Agency research core funding Geography of Slovenia (P6-0101) and Infrastructure Programme (I0-0031). The work of AMP was supported by the Slovenian Research Agency project J1-9185 and research core funding Paleontology and Sedimentary Geology (P1-0008).

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

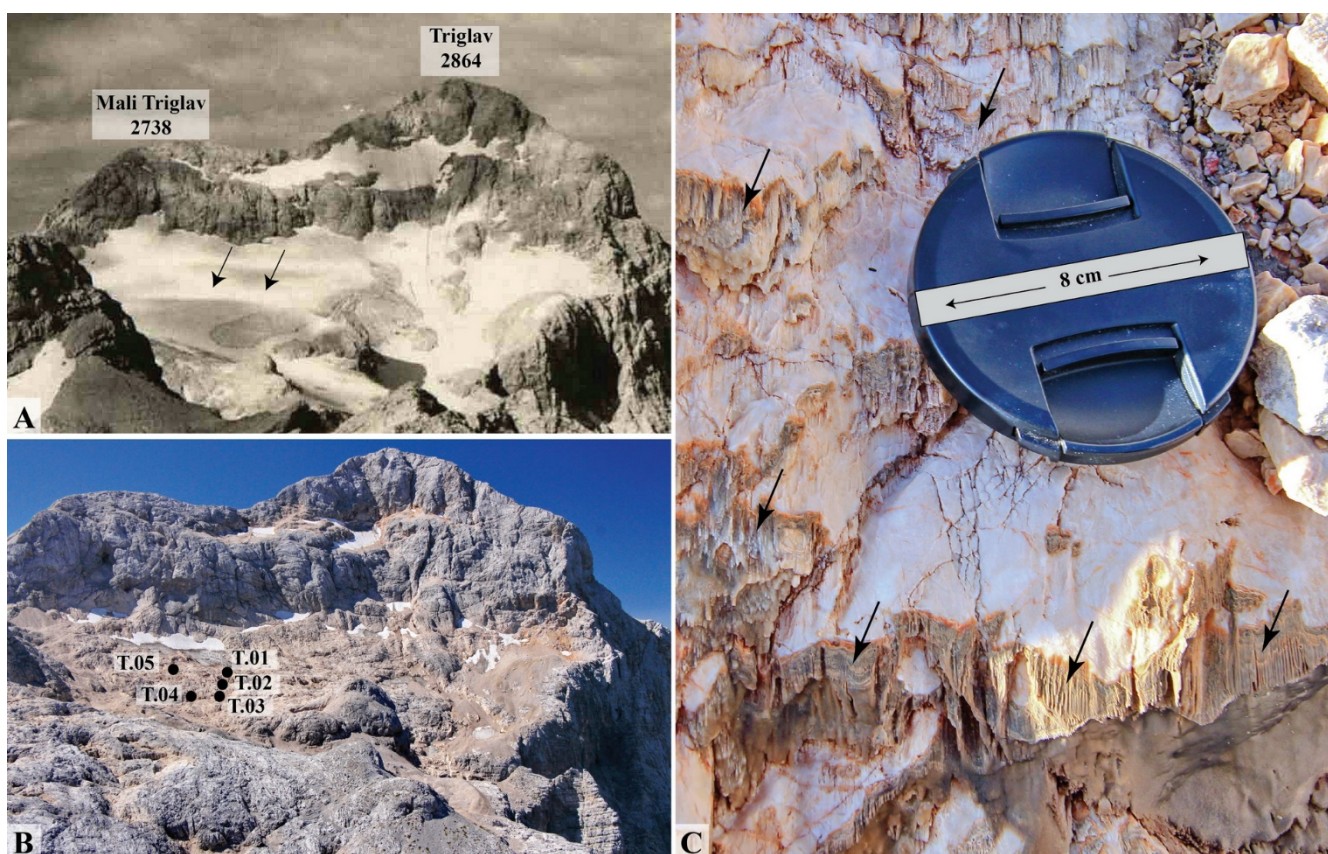

**Figure 1: Mount Triglav, the Triglav Glacier and the sampling location of subglacial carbonates relative to the years 1932 (A; arrows) and 2017 (B; dots). Close up of the exposure of subglacial carbonate (C); arrows point to fluted subglacial carbonate, attached to the lee side of the Upper Triassic limestone. Note the colour-change between the bedrock surfaces: greys (longer**
**exposure) versus beige & brown on the recently exposed surfaces. This has often been explained by microbial activity (Dias et al., 2018). Photo courtesy of (A) Janko Skerlep (© ZRC SAZU Anton Melik Geographical Institute archive), (B) Miha Pavšek and (C) Matej Lipar.**

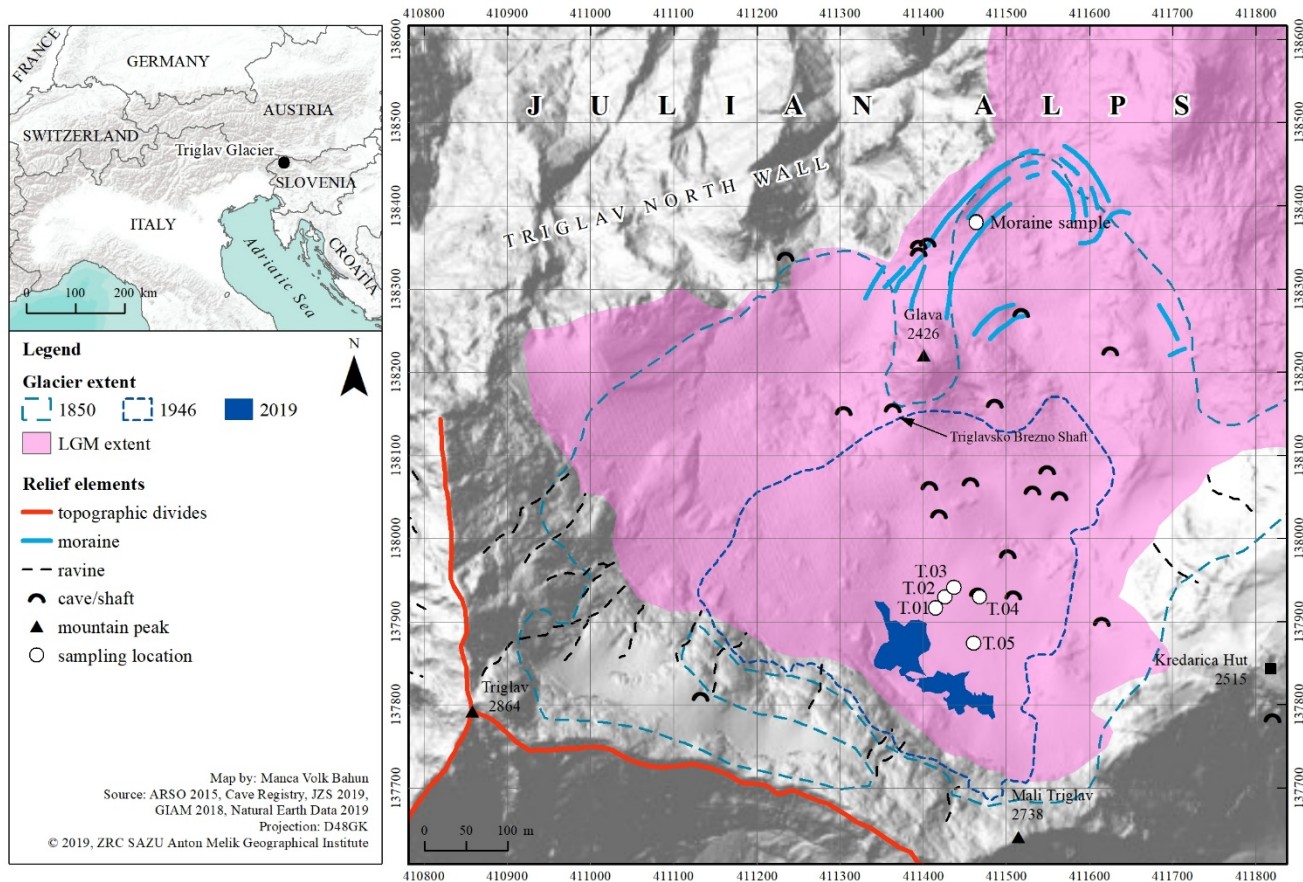

**Figure 2: Locality map of the Triglav Glacier, the sampling sites discussed in the text, and general geomorphology.**


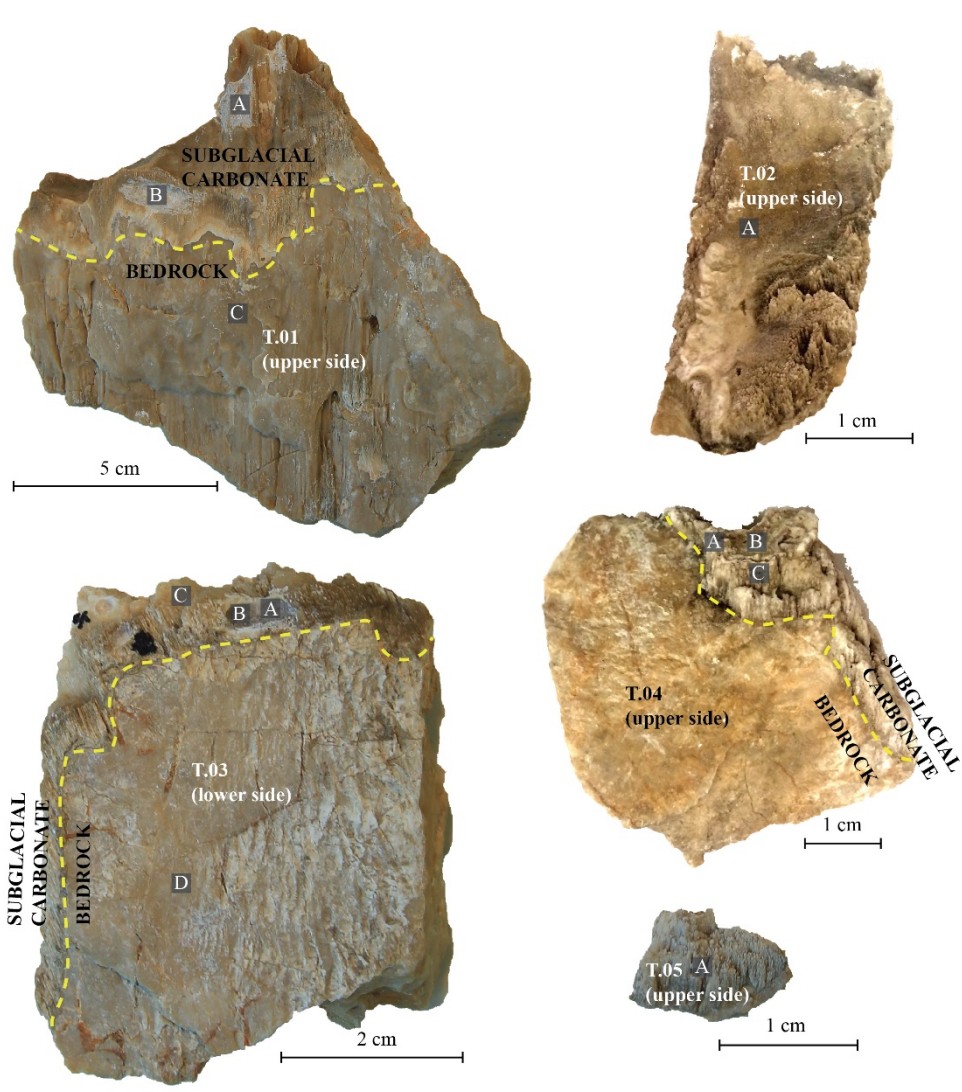

| Sample | Bedrock / subglacial carbonate | XRD mineralogy | δ¹³C (‰ PDB) | δ¹⁸O (‰ PDB) | U-Th ages (ka) |
|---|---|---|---|---|---|
| T.01 A | subglacial carbonate | calcite | 2.41 | -2.84 | 18.45 |
| T.01 B | subglacial carbonate | calcite, aragonite | | | |
| T.01 C | bedrock | calcite | | | |
| T.02 A | subglacial carbonate | calcite | | | |
| T.03 A | subglacial carbonate | calcite, aragonite | 1.35 | -4.71 | 12.72 |
| T.03 B | subglacial carbonate | calcite, aragonite | 0.51 | -5.45 | 3.85 |
| T.03 C | subglacial carbonate | calcite | | | 1.96 |
| T.03 D | bedrock | calcite | | | |
| T.04 A | subglacial carbonate | calcite | 1.50 | -3.96 | |
| T.04 B | subglacial carbonate | calcite, aragonite | | | |
| T.04 C | subglacial carbonate | calcite | | | |
| T.05 A | subglacial carbonate | calcite | 0.98 | -4.63 | 23.62 |
| Average | | | 1.35 | -4.32 | |

**Figure 3: Subglacial carbonate samples used for this study. Upper and lower sides refer to the surface (facing open air before being collected) and underside of the carbonate samples, respectively. The depths were measured relative to the starting drilling point of each sample as they are shown in this figure (lower side perspective for sample T.03 and upper side perspective for all the other samples); only the T.03 sample had precipitates thick enough that allowed to obtain several dates in three different depths.**


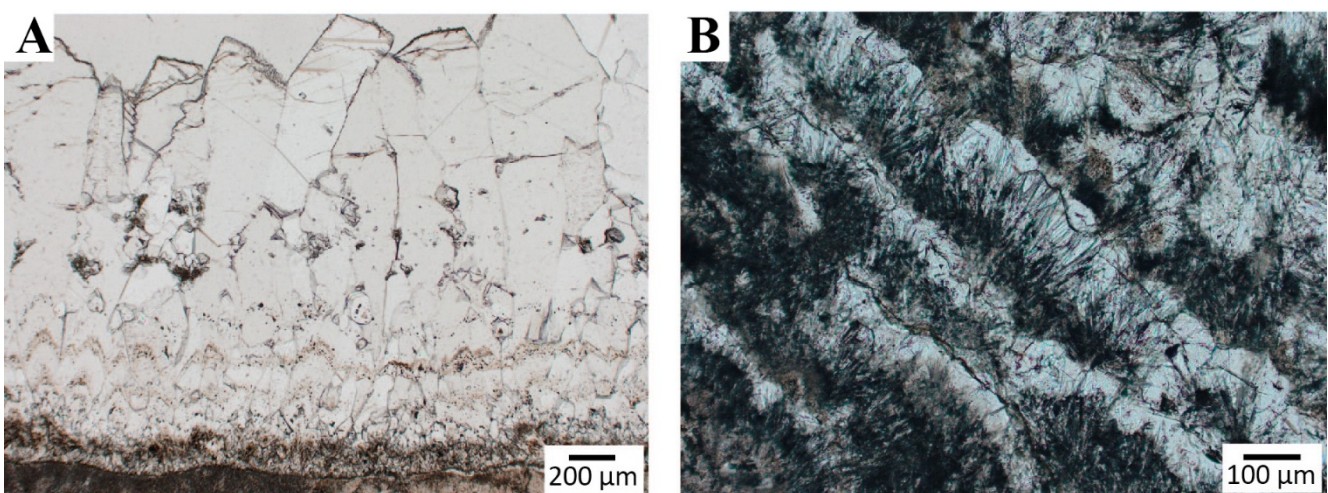

**Figure 4: Petrography of the carbonate crusts. a) Primary columnar calcite crystals growing perpendicularly to the substrate on the steeper side of the bedrock irregularity.; b) alternation of aragonite (fibrous) and calcite crystals forming layered textures. The base of the aragonite fans of crystals nucleates in dark micritic aggregates. Both images were taken under plane polarised light.**





| Surface denudation in karst areas (mm/a) | | | |
|---|---|---|---|
| **Location / Source** | **From** | **To** | **Precipitation (mm)** |
| Dachstein, Austria, 1700 - 1800 m a.s.l. (Bauer, 1964) | 0.015 | 0.020 | 1500 |
| Malham area, NW England, 400 - 500 m a.s.l. (Sweeting, 1964) | 0.040 | / | 1500 |
| Western Julian Alps, 2000 - 2100 m a.s.l. (Kunaver, 1978) | 0.094 | / | 3500 |
| Average for Slovenia, 0 - 2864 m a.s.l. (Gams, 2004) | 0.020 | 0.100 | < 900 - > 3200 |
| Northern Calcareous Alps, Austria, 1500 - 2277 m a.s.l. (Plan, 2005) | 0.011 | 0.048 | 1377 |
| Classical Karst area and Istrian Karst, 0 - 440 m a.s.l. (Furlani et al., 2009) | 0.009 | 0.140 | 1015 - 1341 |
| Tietar Valley, central Spain, 427 m a.s. l. (Krklec et al., 2016) | 0.018 | 0.025 | 797 |
| | | | |
| | | | **Average (rounded)** |
| Min | 0.009 | 0.020 | **0.01** |
| Avg | 0.0296 | 0.0666 | **0.05** |
| Max | 0.094 | 0.140 | **0.1** |

| Calculation of subglacial carbonate existence | | | |
|---|---|---|---|
| Thickness (mm) | Existence - max (years) | Existence - avg (years) | Existence - min (years) |
| 0.1 | 10 | 2 | 1 |
| 0.2 | 20 | 4 | 2 |
| 0.3 | 30 | 6 | 3 |
| 0.4 | 40 | 8 | 4 |
| 0.5 | 50 | 10 | 5 |
| 1 | 100 | 20 | 10 |
| 1.5 | 150 | 30 | 15 |
| 2 | 200 | 40 | 20 |
| 2.5 | 250 | 50 | 25 |
| 3 | 300 | 60 | 30 |
| 4 | 400 | 80 | 40 |
| 5 | 500 | 100 | 50 |

**Table 1: Calculation of subglacial carbonate existence under meteoric environment based on its thickness, years of exposure and denudation rate.**