# Peer review of "Subglacial carbonate deposits as a potential proxy for a glacier's former presence"

_The Cryosphere, 2020_

## Referee Comment (RC1) · Renato R. Colucci (Referee) · 19 Jun 2020

GENERAL COMMENTS Dear Editor, I've read the manuscript "Subglacial carbonate deposits as a potential proxy for glacier's existence" by Matej Liapr et al., submitted to the Journal The Cryosphere. This is a very important topic and I congratulate the authors for having had the idea of doing this research. The vision of a little ice age glacier-extension in the Alps as a short-time (few centuries) events, persisted for decades, but now researchers and climatologists, thanks to a continuously increasing number of datasets and proxies, are approaching a different perspective of alpine glaciation size during the Holocene. This is certainly a topic which will grow attention in the next years. I'm excited about future results which are really important especially for the understanding of the magnitude of present unprecedented abrupt glacier-decay

observed in the Alps due to Anthropogenic global warming. Nevertheless, If I'm very happy with the topic, on the other hand, I guess this manuscript deserves more attention to how things are presented. The lack of discussion is evident at times, as well as a deeper analysis of the paleoclimate significance of the results with the huge amount of existing literature about LGM and Holocene glaciation. The English is good, although sometimes it needs a few adjustments

SPECIFIC COMMENTS

L26-27 When referring to the definition of glacieret a good reference is also the Unesco glossary of glacier mass balance and related terms by Cogley et al. available at this link https://unesdoc.unesco.org/ark:/48223/pf0000192525 Nevertheless, Serrano et al. 2011 gave a very interesting view of such minor ice bodies discussing their evolution from a disintegrating glacier or in areas where nival processes are dominant. To me, it would be important to add also this view in the introduction.

L 29-30 Pay attention, reference Bahr and Radić, 2012 should be highlighted after the sentence "occupy a significant volume fraction at regional scales" and not after "and are thus an important target for palaeoclimate studies"... they never stated this. Anyway, the sentence is overall questionable because the maximum size of the Triglav glacier during the Holocene was much larger than the size of a glacieret. I suggest rewriting the sentence in order to clarify this important aspect

Line 39-43 Subglacial carbonate crusts are also reported in the European Julian Alps by Colucci, 2016 (ESPL page 1232, Geomorphic influence on small glacier response to post-Little Ice Age climate warming: Julian Alps, Europe)

Line 51 You should be consistent with the given definition of glacieret, representing the actual state of this ice body. Honestly, as mentioned above, I would prefer the definition given by Serrano et al., 2011 and classify this ice body as a "glacial ice patch", meaning that it is actually an ice patch (no more than 2-3 m thick), residual ice body of a recently flowing glacier.

Lines 84 and 94 Add space after "(sub)"

Line 127-133 I would suggest replacing here the term "glacieret" with "glacier" espe-
cially because when referring to the further chapter at line 135-138 is correctly stated
that carbonate deposition resemble flow direction of the glacier and precipitation was
strongly influenced by the mechanical force of the ice movement. Please, give clues
about the location and number/name of such younger dated samples and of all the
cited samples. It is important for the reader to understand where each dated sample
is located in the surrounding s of the present ice patch which is a non-moving ice/firn
mass. Nevertheless, after reading all the manuscript, I think this sub-chapter is rather
unuseful and might be deleted because they are better presented and discussed in
chapter 4

Line 174-175 In that paper Resfinder et al. stressed the fact that carbonate crusts
were preserved by very cold and dry Arctic climate, which is really not the case of the
Triglav cirque in the last century or during any possible period of absence or almost-
complete-absence of an ice body. This is particularly true when looking at Mean Annual
Precipitation (MAP) of 2600 mm w.e. recorded at Kredarica.

Line 182 I'm wondering if this could be entirely correct. At the LGM peak,
Kuhleman et al2008 suggested a lowering of summer temperature of about 9-
11 °C in the southeastern Alps. This roughly means that at Kredarica (2514
m), where the present summer temperature (1981—2010) is around +6.0 °C
(http://meteo.arso.gov.si/uploads/probase/www/climate/table/sl/by_location/kredarica/climate-
normals_81-10_Kredarica.pdf) and considering the recent warming in the area
calculated in the southeastern Alps in +1.7°C since the end of the Little Ice Age by
Colucci & Guglielmin 2015, should have been roughly between -4.7 and -6.7. These
characteristics lead to the existence at that location of a cold base glacier, instead
of a temperate glacier. Nevertheless, given dates at 23.62 ka, 18.45 ka, and 12.72
ka suggest some interesting speculation. As stated by Monegato et al., 2017 in
the paper "The Alpine LGM in the boreal ice-sheets game" published in Scientific

[Figure]

Reports (https://www.nature.com/articles/s41598-017-02148-7), the LGM seems to be characterized by 2 main peaks with a withdrawal phase between 23.9 and 23.0 ka when tha Garda glacier retreated, which fit rather well with the 23-62 ka date given in this work. The final collapse of the Garda glacier occurred around 17.7-17.3 ka but soon before there was a progressive stacking of moraines related to the retreat and lowering of the ice surface which seems to fit well with the 18.45 ka dating. Both events could represent the occurrence of "some" subglacial water at the glacier-bed. Finally, I have no problem considering that during the Younger Dryas phase there was certainly a large amount of free water flowing at the glacier-bed. Small scale Detailed paleoclimate conditions in the Alps are still an open question, and uncertainties are evident especially in the eastern Alps when looking at bias between the modeled MIS2 stage ice extent (Seguinot et al., 2018) and geomorphological reconstruction (Ehlers et al., 2011), but the discussion could be expanded in such a way.

Lines 195-200 this is too speculative in my opinion. There is no evidence at present of cave ice older than roughly 10 ka at least in Europe, on my best knowledge. I would be more cautious in this manuscript deleting "perhaps even Last Glacial Maximum times"

Line 236 I would prefer "ablation" instead of "melting"

Line 241 I might agree with what it is here stated, but as a possible cause I would also cite the important work of Painter et al., 2013 (https://www.pnas.org/content/110/38/15216)

Line 242-248 This part is too hasty, although crucial in the discussion, and should be more deeply investigated and discussed. For instance, has been shown as the retreat of Triglav glacier since the LIA in the last century has been more evident than in other sectors of the Julian Alps where other glaciers existed. This is the case of Canin-Kanin or Montasio West glaciers which are lower in elevation than the Triglav glacier. The Montasio West is still classified as a moving glacier with dynamics due to internal deformation. The reason why Canin-Kanin and Zeleni Sneg (the largest LIA

glaciers in the Julian Alps) had different fates in terms of shrinking velocity and a 200 m difference in the Equilibrium Line Altitude (ELA) has been justified by Colucci 2016 to a large difference in Mean Annual Precipitation (in the Triglav area precipitation are roughly 60% of that in Canin-Kanin) and potential annual solar radiation for the glaciers differed by about 7%. In this view Triglav glacier generally has higher sensitivity to summer temperature while Canin-Kanin lies in a more "maritime" environment and is more sensitive in changes of winter precipitation. I guess a discussion in terms of variability of these two parameters during the Holocene and/or in the Lateglacial period would improve this part of the manuscript. The literature is quite abundant on this topic.

FIGURES Figure 3 Is not adding anything crucial to the study area or the manuscript itself. It Maybe could be deleted and glacier outlines drawn in figure 2. Instead, Figure S2 could be added in the main article as Figure 3, maybe highlighting with arrows and numbers the location of the samples because in the main manuscript a picture of the study area with a view of the present state of the ice patch is missing

Suppl. Material Figure S3 . . . it would be useful to add a number of samples together with arrows Figure S5 . . . not clear which of the 3 caves is the Tiglavski Brezno Shaft. . . Fonts and size are not the best, please improve the size and the visibility of the text Figure S6 . . . I would change "melting season" with "ablation season". More, please give clues about the moving average used (how many years ?! It is centered ?) Figure S7 . . . Besides linear regression, I would also add a moving average which probably better highlights variability along about the last 170 years. These are indeed very interesting data, I'm asking my self if they are available in some repository to the scientific community.

Pointing out once more the importance of this in a way pioneering work, I think it deserves publication but after careful updatings and insights Dr. Renato R. Colucci, PhD

---

## Author Comment (AC1) · 11 Jul 2020

**Response to Dr. Renato R. Colucci (Reviewer #1) to manuscript TC-2020-82**

*Italic:* Referee comments

**Bold:** Authors comments

Red: Selected changes in the manuscript (note, not all changes are shown here, but will be submitted as revised manuscript with track-changes)

SPECIFIC COMMENTS

Referee:

*L26-27 When referring to the definition of glacieret a good reference is also the Unesco glossary of glacier mass balance and related terms by Cogley et al. available at this link https://unesdoc.unesco.org/ark:/48223/pf0000192525 Nevertheless, Serrano et al. 2011 gave a very interesting view of such minor ice bodies discussing their evolution from a disintegrating glacier or in areas where nival processes are dominant. To me, it would be important to add also this view in the introduction.*

Authors:

**The terminology is indeed vague when it comes to exact definition of the glacieret, so the definitions and discussions of Cogley et al. and Serrano et al. are now added in the text, and readers are referred to these two references for further reading.**

…glacierets are defined as a type of miniature (typically less than 0.25 km$^2$) glaciers or ice masses of any shape persisting for at least two consecutive years (Cogley et al., 2011; Kumar, 2011). Serrano et al. (2011) differ them from ice patches in terms that glacierets are "the product of larger ancient glaciers, still showing motion or ice deformation, although both very low. They have a glacial origin, glacial ice and are never generated by new snow accumulation", whilst ice patches are "ice bodies without movement by flow or internal motion". Despite their small size, glacierets occupy a significant volume fraction at regional scales (Bahr and Radić, 2012), and can therefore be considered as an important target for palaeoclimate studies. Accordingly, many present-day glacierets are closely monitored and studied (Gądek and Kotyrba, 2003; Grunewald and Scheithauer, 2010; Gabrovec et al., 2014; Colucci and Žebre, 2016), but the peculiarity of current global climate change requires more evidence from different proxies and from further in the past when current ice patches and glacierets were still glaciers…

Referee:

*L 29-30 Pay attention, reference Bahr and Radi´c, 2012 should be highlighted after the sentence "occupy a significant volume fraction at regional scales" and not after "and are thus an important target for palaeoclimate studies". . . they never stated this. Anyway, the sentence is overall questionable because the maximum size of the Triglav glacier during the Holocene was much larger than the size of a glacieret. I suggest rewriting the sentence in order to clarify this important aspect.*

Authors:

**We corrected the citation order, and also made it clearer that Triglav Glacier (Glacier with the capital as it is its official name) is now ice patch, but used to be glacier (and glacieret) in the past.**

Referee:

*Line 39-43 Subglacial carbonate crusts are also reported in the European Julian Alps by Colucci, 2016 (ESPL page 1232, Geomorphic influence on small glacier response to post-Little Ice Age climate warming: Julian Alps, Europe).*

Authors:

**The reference was added to the text.**

Referee:

*Line 51 You should be consistent with the given definition of glacieret, representing the actual state of this ice body. Honestly, as mentioned above, I would prefer the definition given by Serrano et al., 2011 and classify this ice body as a "glacial ice patch", meaning that it is actually an ice patch (no more than 2-3 m thick), residual ice body of a recently flowing glacier.*

Authors:

**We largely reworked the whole introduction chapter so it is clearly stated that Triglav Glacier is at present a "glacial ice patch" (and was also recently a "glacieret", and less recently a "glacier").**

Referee:

*Lines 84 and 94 Add space after "(sub)"*

Authors:

**Corrected.**

Referee:

*Line 127-133 I would suggest replacing here the term "glacieret" with "glacier" especially because when referring to the further chapter at line 135-138 is correctly stated that carbonate deposition resemble flow direction of the glacier and precipitation was strongly influenced by the mechanical force of the ice movement. Please, give clues about the location and number/name of such younger dated samples and of all the cited samples. It is important for the reader to understand where each dated sample is located in the surrounding s of the present ice patch which is a non-moving ice/firn mass. Nevertheless, after reading all the manuscript, I think this sub-chapter is rather unuseful and might be deleted because they are better presented and discussed in chapter 4.*

Authors:

**The term "glacieret" was replaced with "glacier". We also made it clearer in the Methodology chapter how many samples were collected and referred a reader to the location figures** (Five subglacial carbonates were collected 50 m to 100 m from the current ice patch (Fig. 2).)**. Since this chapter is strictly representing the results only (as the chapters before this), followed by a discussion chapter, we still left the chapter included in the manuscript.**

Referee:

*Line 174-175 In that paper Resfinder et al. stressed the fact that carbonate crusts were preserved by very cold and dry Arctic climate, which is really not the case of the Triglav cirque in the last century or during any possible period of absence or almostcomplete- absence of an ice body. This is particularly true when looking at Mean Annual Precipitation (MAP) of 2600 mm w.e. recorded at Kredarica.*

Authors:

**This is a helpful additional argument for the continuously existing Triglav Glacier during the Holocene, and we added this part to the chapter 5.**

…whilst Refsnider et al. (2012) discussed the preservation of the subglacial carbonates by very cold and dry Arctic climate, this cannot be the case of the cirque of Triglav Glacier in the last century or during any possible period of absence of an ice body, evident by mean annual precipitation at Kredarica hut (2515 m a.s.l.) (Slovenian Environment Agency, 2020) (see Fig. 2 for location)….

Referee:

*Line 182 I'm wondering if this could be entirely correct. At the LGM peak, Kuhleman et al2008 suggested a lowering of summer temperature of about 9-11°C in the southeastern Alps. This roughly means that at Kredarica (2514 m), where the present summer temperature (1981ă˘Ă˘ T2010) is around +6.0 ° C (http://meteo.arso.gov.si/uploads/probase/www/climate/table/sl/by_location/kredarica/climatenormals _81-10_Kredarica.pdf) and considering the recent warming in the area calculated in the southeastern Alps in +1.7 C since the end of the Little Ice Age by Colucci & Guglielmin 2015, should have been roughly between -4.7 and -6.7. These characteristics lead to the existence at that location of a cold base glacier, instead of a temperate glacier. Nevertheless, given dates at 23.62 ka, 18.45 ka, and 12.72 ka suggest some interesting speculation. As stated by Monegato et al., 2017 in the paper "The Alpine LGM in the boreal ice-sheets game" published in Scientific Reports (https://www.nature.com/articles/s41598-017-02148-7), the LGM seems to be characterized by 2 main peaks with a withdrawal phase between 23.9 and 23.0 ka when tha Garda glacier retreated, which fit rather well with the 23-62 ka date given in this work. The final collapse of the Garda glacier occurred around 17.7-17.3 ka but soon before there was a progressive stacking of moraines related to the retreat and lowering of the ice surface which seems to fit well with the 18.45 ka dating. Both events could represent the occurrence of "some" subglacial water at the glacier-bed. Finally, I have no problem considering that during the Younger Dryas phase there was certainly a large amount of free water flowing at the glacier-bed. Small scale Detailed paleoclimate conditions in the Alps are still an open question, and uncertainties are evident especially in the eastern Alps when looking at bias between the modeled MIS2 stage ice extent (Seguinot et al., 2018) and geomorphological reconstruction (Ehlers et al., 2011), but the discussion could be expanded in such a way.*

Authors:

**This discussion was avoided in the first draft due to the greater focus into preservation of subglacial carbonates and the lack of high-number/resolution dates to constrain the individual withdrawal periods during the LGM. Nevertheless, the retreat of the Garda Glacier indeed points out rather interesting correlations to the single ages of the subglacial carbonates of the Triglav Glacier, so we added the proposed discussion in the text.**

…the cold Alpine environment during the glacial period with low biological respiration rates could be indicated in the relatively high δ13C signal, also reported by others (Lemmens et al., 1982; Fairchild and Spiro, 1990; Lyons et al., in press), but the (re)freezing of the subglacial water causes supersaturation with respect to carbonate and the non-equilibrium conditions produced by this process can affect the stable isotopic composition of the subglacial carbonate, usually leading to isotopic enrichment in the carbonate minerals (Clark and Lauriol, 1992; Courty et al., 1994; Lacelle, 2007). In any case, the climate during the precipitation of subglacial carbonate had to be "warm" enough to produce subglacial water at the glacier-bed. Kuhlemann et al. (2008) suggested a lowering of summer temperature of about 9-11°C at the LGM peak, meaning that the summer temperatures at Triglav Glacier should have been around -4.7 to -6.7°C (based on the present summer temperatures at around +6°C (Slovenian Environment Agency, 2020) and additional considering the recent warming of +1.7°C in the area calculated in the southeastern Alps since the end of the Little Ice Age by Colucci and Guglielmin (2015)), which would have been conditions for the cold base glacier and the absence of subglacial water flow. Whilst small scale detailed palaeoclimate conditions in the southeastern Alps are still uncertain, the current U-Th ages of subglacial carbonates within this research fit with the temporary Garda Glacier

withdrawal phases reported by Monegato et al. (2017); 23.62 ka age of subglacial carbonate relates to the withdrawal phase between 23.9 and 23.0 ka, and 18.45 ka could relate to the glacier retreat from around 19.7 to 18.6 ka or even the final collapse of the glacier around 17.7 to 17.3 ka; both time periods could relate to commencement of subglacial water flow also at the Triglav Glacier and consequently the precipitation of subglacial carbonates. In addition, the 12.72 ka age of Younger Dryas would predate the period of maximum cooling between ca. 11-10 ka (Mathewes, 1993; Renssen and Isarin, 1997)…

Referee:

*Lines 195-200 this is too speculative in my opinion. There is no evidence at present of cave ice older than roughly 10 ka at least in Europe, on my best knowledge. I would be more cautious in this manuscript deleting "perhaps even Last Glacial Maximum times".*

Authors:

**We deleted this part.**

Referee:

*Line 236 I would prefer "ablation" instead of "melting".*

Authors:

**We changed the words accordingly.**

Referee:

*Line 241 I might agree with what it is here stated, but as a possible cause I would also cite the important work of Painter et al., 2013 (https://www.pnas.org/content/110/38/15216).*

Authors:

**We included this work (and also added it in the chapter 7).**

Referee:

*Line 242-248 This part is too hasty, although crucial in the discussion, and should be more deeply investigated and discussed. For instance, has been shown as the retreat of Triglav glacier since the LIA in the last century has been more evident than in other sectors of the Julian Alps where other glaciers existed. This is the case of Canin-Kanin or Montasio West glaciers which are lower in elevation than the Triglav glacier. The Montasio West is still classified as a moving glacier with dynamics due to internal deformation. The reason why Canin-Kanin and Zeleni Sneg (the largest LIA glaciers in the Julian Alps) had different fates in terms of shrinking velocity and a 200 m difference in the Equilibrium Line Altitude (ELA) has been justified by Colucci 2016 to a large difference in Mean Annual Precipitation (in the Triglav area precipitation are roughly 60% of that in Canin-Kanin) and potential annual solar radiation for the glaciers differed by about 7%. In this view Triglav glacier generally has higher sensitivity to summer temperature while Canin-Kanin lies in a more "maritime" environment and is more sensitive in changes of winter precipitation. I guess a discussion in terms of variability of these two parameters during the Holocene and/or in the Lateglacial period would improve this part of the manuscript. The literature is quite abundant on this topic.*

Authors:

**We emphasised now the geomorphological peculiarity of small glaciers which influence their dynamics and included the Canin example. Nevertheless, the aim of the paper is to demonstrate the value of subglacial carbonates which offer possible indications of on-going persistence of the Triglav Glacier since the LGM, which based on geomorphological predispositions would be**

**amongst the first ones to disappear, despite its higher altitude, and therefore the most appropriate one to relate this to other small glaciers. However, as stated in the paper, high-resolution analyses of subglacial carbonates need to follow to justify further environmental discussions.**

…on the other hand, further comparable research on various small glaciers is needed to generalise the palaeoclimatic data due to geomorphological peculiarity of glacier regions; for example, even though the resilience of the Triglav Glacier until present can be emphasised due to the above described regional components, the retreat of the Triglav Glacier has been more evident than the retreat of the Canin Glacier (Italy), Montasio West Glacier (Italy) and Skuta Glacier (Slovenia) (all in southeastern Alps), which are all lower in elevation than the Triglav Glacier, but large difference in mean annual precipitation (e.g., in the Triglav area water equivalent precipitation (2071 mm) is 62% of that in Canin area (3335 mm)) and potential annual solar radiation play a major role for differences in glaciers' dynamics (Colucci, 2016)…

FIGURES

Referee:

*Figure 3 Is not adding anything crucial to the study area or the manuscript itself. It Maybe could be deleted and glacier outlines drawn in figure 2. Instead, Figure S2 could be added in the main article as Figure 3, maybe highlighting with arrows and numbers the location of the samples because in the main manuscript a picture of the study area with a view of the present state of the ice patch is missing.*

Authors:

**Figure 3 was deleted. We added the locations of samples in Figure 1 B, which is showing the area of the present state of the ice patch. Figure S2 and S3 are therefore also not needed as they are similar to Figure 1 B and were therefore deleted. The ice outline from previous Figure 3 was added (only 1946 year) to the Figure 2, whilst additional outlines can still be traced in supplementary material. All of the figures have now text changed into Times New Roman and in places enlarged to be more visible.**

[Figure]

**Fig. 1**

[Figure]

**Fig. 2**

Referee:

*Suppl. Material Figure S3 . . . it would be useful to add a number of samples together with arrows Figure S5 . . . not clear which of the 3 caves is the Tiglavski Brezno Shaft. . . Fonts and size are not the best, please improve the size and the visibility of the text Figure S6 . . . I would change "melting season" with "ablation season". More, please give clues about the moving average used (how many years ?! It is centered ?) Figure S7 . . . Besides linear regression, I would also add a moving average which probably better highlights variability along about the last 170 years. These are indeed very interesting data, I'm asking my self if they are available in some repository to the scientific community.*

Authors:

**We made on all the figure maps clear now which is the Triglavski Brezno Shaft, and also improved fonts and size. Figure S6 is updated, also Figure S7; we included now the 20-year uncentered moving average for both graphs. The raw data of Figure S6 is available through Slovenian Environment Agency site (https://www.arso.gov.si/en/), and a number of additional raw data of the Triglav Glacier are accessible on the dedicated page of the Slovenian Environment Agency (http://kazalci.arso.gov.si/en/content/triglav-glacier). The original reconstruction data of Figure S7 was primarily published in Gabrovec et al. (2014) and the reader is referred to this reference (as the caption states as well); the reconstruction of the highest seasonal snow height is not the main topic of the manuscript, however, due to its recognition as a valuable data, the process has been started to build a repository upon the referenced (Gabrovec et al., 2014) work.**

[Figure]

[Figure]

**Again, we thank dr. Colucci for the fair and constructive review.**

**Authors.**

---

## Referee Comment (RC2) · Bernard Hallet (Referee) · 21 Sep 2020

**Review of Ms tc-2020-82 subglacial carbonate precipitates**

**Notes**

This is a fine paper, clearly presented and well illustrated, but only with skeletal captions that do not do justice to the figures. The paper describes a previously unreported occurrence of subglacial precipitates, and reports considerable data including ages that are much older than expected. The authors appropriately stress the significance of their findings; subglacial precipitates are a novel palaeo-environmental proxy and a research subject well worth further research. Their findings suggest new evidence that highlight the extraordinary nature of the current global warming.

One important improvement would be to add credibility to the ages reported by providing more explicit details about the impact on the calculated age of the initial content of Thorium 230 in the precipitate. One effective way of doing this is in table form much as that shown below from Fitzpatrick, J. J., Muhs, D. R., & Jull, A. J. T. (1990). (Saline minerals in the Lewis Cliff ice tongue, Buckley Island quadrangle, Antarctica. *Contributions to Antarctic Research I*, *50*, 57-69). In particular, for 230/232 values of 4, for example, the age could be as much as 40% younger than the age calculated that does not assume there is any 230 initially. Also, the text should reflect as accurately as possible the corresponding large uncertainties.

TABLE 3. Uranium and Thorium Concentrations, Isotopic Activity Ratios, Uranium Series Ages, and Radiocarbon Ages of Antarctic Saline Minerals

| Site | Mineralogy | U, ppm | Th, ppm | $^{234}U/^{238}U$ | $^{230}Th/^{232}Th$ Activity Ratios | $^{230}Th/^{234}U$ | Apparent Age* (ka) Using Correction for Initial $^{230}Th/^{232}Th$ Ratios | | | | | $^{14}C$ Age, years |
|---|---|---|---|---|---|---|---|---|---|---|---|---|
| | | | | | | | 0 | 0.5 | 1.0 | 1.5 | 2.0 | |
| 1 | nahcolite | 0.73 ± 0.01 | 0.004 ± 0.001 | 5.04 ± 0.04 | 185 ± 25 | 0.066 ± 0.001 | 7.4 ± 0.1 | NA | NA | NA | NA | 24,560 ± 420 |
| 2a | nahcolite | 0.88 ± 0.01 | 0.023 ± 0.002 | 4.19 ± 0.03 | 28 ± 2 | 0.055 ± 0.001 | 6.1 ± 0.1 | NA | NA | NA | NA | 21,410 ± 315 |
| 2b | nahcolite | 0.614 ± 0.008 | 0.122 ± 0.003 | 3.26 ± 0.03 | 4.0 ± 1 | 0.078 ± 0.001 | 8.8 ± 0.2 | 7.7 | 6.6 | 5.5 | 4.4 | 34,470 ± 710 |
| 2c | trona | 0.299 ± 0.003 | 0.095 ± 0.002 | 2.35 ± 0.02 | 1.30 ± 0.04 | 0.074 ± 0.002 | 8.3 ± 0.2 | 5.2 | 2.0 | 0 | 0 | postbomb |
| 2d | nahcolite + borax | 0.121 ± 0.003 | 0.046 ± 0.005 | 3.24 ± 0.09 | 14 ± 2 | 0.55 ± 0.01 | 78 ± 3 | 76 | 75 | 73 | 71 | 9,960 ± 100 (nahcolite only) |
| 2d | nahcolite (matrix) | 0.057 ± 0.001 | 0.020 ± 0.001 | 3.93 ± 0.07 | 30 ± 1 | 0.85 ± 0.01 | 150 ± 4 | NA | NA | NA | NA | 37,730 ± 1060 |

NA, not applicable because measured $^{230}Th/^{232}Th$ is ≥ 20.

*Calculated from corrected $^{230}Th/^{234}U$ ratios using the following equation: $^{230}Th/^{234}U_{corrected} = ^{230}Th/^{234}U_{measured} - [^{230}Th/^{232}Th_{initial} (^{232}Th/^{234}U_{measured}) \exp(-\lambda_{230}t)]$, where $\lambda_{230}$ is the decay constant of $^{230}Th$ and $t$ is the age of the sample. The equation is solved by successive approximations.

The authors may also wish to consider leveraging the limelight of Ötzi, the Iceage Man, and its climate implications, as referenced by Solomina et al (2015) in their supplementary material.

**Specific Comments by line number**

29. …significant volume fraction of what?

41. Also reported from the southern tip of S. America (Tierra del Fuego, Personal communication, Rabassa), New Guinea (Peterson and Moresby, 1979), and from sites where they formed under LGM ice.

56-57. The units, kg/m$^2$/yr, seem unusual. Why not report ice thinning rate in m/yr, or the rate of increase of exposed bedrock, m$^2$/yr? This rate must be averaged over a certain area, but what is it? This reference, Gabrovec et al., 2014, does not help; it is incomplete and insufficient.

Fig. 1 caption should be more informative, explaining to unfamiliar readers
- what is what (bedrock vs. precipitate)?
- the orientation of surface imaged relative to horizontal and to the former sliding direction
- the morphology of the precipitates

Fig. 2.  What are is the pink areas?  Replace these terms in legend; in English they are incorrect or awkward.

**From:**

**Relief types**
Erosional topography
Depositional topography
Periglacial topography

**Relief Shapes**
Main ridge

**To:**

**Terrain types**
Erosional surfaces
Depositional surfaces
Periglacial terrain

**Relief Elements**
Topographic divides

60.  Replace "…the recently exposed subglacial carbonate deposits due to glacier retreat." by "…the subglacial carbonate deposits recently exposed by glacier retreat."

78.  (…targeting carbonate) cement ought to be replaced by the more appropriate word, precipitate

103. Replace "They are fluted and furrowed crust-like deposits characterized by brownish, greyish or yellowish colour." by "The fluted and furrowed crust-like deposits are brownish, greyish or yellowish in colour."

115-6. "Depending on the angle of the lee side of bedrock protuberances, columnar calcite crystals grow either perpendicularly to the host rock (Fig. 5a) or with a lower angle, generally oriented downslope…"  Replace "angle" by "inclination" or "slope".  It would be good to explain how the crystal orientation varies with bedrock surface inclination less ambiguously.  For example, do vertical crystals grow perpendicular to the rock when bedrock surface is near vertical or near horizontal?  Is there a relation between the crystal orientation and the former glacier sliding direction?

124. Replace is by are.

127.  For the isotopic ratios, the ranges should be included in the text, as well as averages.  The reader should not have to look up the supplement for this basic information.

129-134.  Briefly explain what ages you expected.  Weere the "two U-Th ages of stratigraphically younger cement" obtained from the same sample?   If any of the thin sections are from this sample, you should mention it in the text, and help understand the stratigraphic setting of these younger deposits. Why would the former glacier be thick, and what do you mean by thick?

147. I'd replace low supersaturated solutions by slightly supersaturated solutions

148-9.  Replace "high Mg/Ca ratio in the water partially could be the trigger for the precipitation of aragonite" by "high Mg/Ca ratios in the water partially be responsible for the precipitation of aragonite".

150-1.  Recast sentence to avoid circular logic.

152.  I suggest replacing "pose a challenge to determine" by "raise the difficult question of"

159-160.  What is it about the moment that matters in the following "… factors such as the percentage of initial aragonite and the moment of the aragonite to calcite transformation…the possible additional redistribution of Th …. or the degree of opening of the system"?  How about this rewording: "… factors such as the initial relative amount of aragonite and the timing of its transformation to calcite …the possible additional redistribution of Th …. and the extent of chemical exchange with widespread subglacial meltwater"?

162-4.  Suggested edit from:
"Based on the U concentration in samples within this study (in ppm; Supp. Table S2), it is notable that the youngest sample (2ka; T.03_b1) has 1.77 ppm of U concentration, whilst two of the old samples (LGM and YD; T.01_a1 and T.03_a1) have around 0.41 and 0.46 ppm of U concentration, respectively"
To
"It is notable that the U concentration  (in ppm; Supp. Table S2) in the youngest sample (2ka; T.03_b1) within this study is 1.77 ppm, whereas it is around 0.41 and 0.46 ppm in two of the old samples,  T.01_a1 and T.03_a1, respectively LGM and YD."

165. Replace contrary by to the contrary

166 & 168. Replace In case of the first possibility…by Assuming the first possibility…The same goes for line 168.

Figure 4:
An informative caption is needed for this important figure.  The labels are ambiguous. I assume, but am uncertain, that upper and lower "sides" refers to the surface (facing open air before being collected) and underside of the carbonate samples.  In any case, how is depth measured?  Is it relative to the surface or to the underside?  For T.03, does the "lower side" include limestone bedrock as well as subglacial precipitate?  It would be helpful for the reader to indicate clearly in

words or graphics how the ages vary with stratigraphy. Perhaps, this could be dome easily by providing the ages that correspond to the depths written on figure.  Is this the only sample for which several dates have been obtained?

192.  $\delta$18O differences of a few per mill in the carbonate precipitate can also arise due to variations in subglacial hydrology shifting from closed to open geochemical systems (see Hanshaw & Hallet)

197. Replace "lighter in deuterium compared to the Triglav…" by "lighter in deuterium than those from Triglav…"

199. The last clause (constraining the implications that the Triglav Glacier was constant during the Holocene.) does not follow logically from the preceding text.  Clarify or delete it.

203.  I am unsure of your intent with the leading sentence of this section.  If it is consistent with the heading, I would suggest this revision: The LGM and YD ages are the first physical evidence that Triglav Glacier persisted through the Holocene to the present day (Solomina et al., 2015).  If your intent is different, describe it clearly.

210.  I suggest replacing "not being documented in the literature" by "not having been reported in the literature"

214.  This paragraph ought to be updated in view of more recent work reporting even slower denudation (e.g. Steinemann et al., 2020), but still supports the contention that the 5mm crust would have weathered off during the HCO if exposed to the elements.

221. Delete the last 3 words because once exposed the subglacial carbonate deposits they cannot be glacially abraded.  Moreover, I would not expect them to be abraded even under the glacier because they form in lee positions where I would expect abrasion to vanish as abrading rock fragments diverge from the bed at sites of subglacial precipitation due to regelation ice growth.

228. Delete "assessment of"

240.  Replace "quick rate of 21st" by "rapid 21$^{st}$"

243.  Replace "show a lower sensitivity to climate fluctuations" by "are less sensitive to climate fluctuations"

245.  This remark about bright limestone substrate reminds me of the photographs, which begs the question: what is responsible for the color difference (greys vs. beige & brown on the more recently exposed bedrock surfaces)?

250.  What do you mean here?  The preliminary data shows a high possibility that subglacial carbonate deposits may endue unprecedented retreat… Might you mean the following? The preliminary data suggest that subglacial carbonate deposits can archive valuable datable records of glacial retreat, including hints that the current and ongoing retreat is unprecedented.

255. Replace observe by determine.

260. Replace considerably fast by relatively high.  This section should also leverage, and be updated by, recent work (Steinemann et al., 2020).

264-5.  Replace "particulates on the present remnants of ice (and possible ice cores, if a glacier has not disappeared completely)" by "material on the small remaining ice masses and, if possible, ice cores)

274.  The conclusion would be stronger and clearer without the first sentence. I would replace it as follows:
Subglacial carbonate deposits recently exposed by the retreating Triglav Glacier contain the first direct evidence of the existence and extent of Triglav Glacier since the Last Glacial Maximum and Younger Dryas.
The deleted sentences should be incorporated in the previous section and clearly explained:"U-Th ages of subglacial carbonate with the combination of aragonite and calcite are regarded as maximum ages as aragonite-to-calcite transformation, evident in fabrics, might have occurred in calcite crystals that could have been falsely considered as primary.

**Missing refs**
Hanshaw, B. B. and B. Hallet. 1978. Oxygen isotope composition of subglacially precipitated
    calcite: possible paleoclimatic implications. Science, 200,1267-1270.
Peterson, J. A., and Moresby J.F. 1979 Subglacial travertine and associated deposits in the
    Carstensz area, Irian Jaya, Republic of Indonesia. Zeitschrift fur Gletscherkunde und
    Glazialgeologie. 15(1), 23-29
Steinemann, O., Ivy-Ochs, S., Grazioli, S., Luetscher, M., Fischer, U. H., Vockenhuber, C., &
    Synal, H. A. 2020. Quantifying glacial erosion on a limestone bed and the relevance for
    landscape development in the Alps. Earth Surface Processes and Landforms, 45(6), 1401-
    1417.

**Supplementary material**

Figure S2:
The caption states "The recently exposed surface with shafts and subglacial carbonate deposits" but does not address the distinct colors, ranging from greys to rusty brown.  Do the color boundaries correspond to, or parallel outlines of glacial extent in Figure S4?  The figure should also show where the subglacial deposits or shafts are.  In fact, what do you mean by shafts?

Figure S8: XRD graphs.  A short caption is needed to explain what these samples are, how they differ from one another, and which are bedrock (if any).

Comments and questions in italics. Figure S9: a) Short columnar calcite crystals alternating with brown micritic bands constitute *the first phase of calcite precipitation on the bedrock*. Plane polarised light (PPL); b) columnar calcite crystals predominantly oriented towards the right (downslope) .*[what is the orientation of the thin section relative to the former sliding direction? Same question for Fig S10*]. The

growth of the crystal on the center crosscut the direction of growth of previous crystals.*[what is the center crosscut?]* Crossed polarised light (XPL)

Figure S11- caption needs a brief explanation of what the figure shows.  What are the various curves?

---

## Editor Comment (EC1) · Chris R. Stokes (Editor) · 29 Sep 2020

I would like to put on record my thanks to the reviewers for their generally very encouraging and constructive comments on this manuscript. They identify a large number of relatively minor issues which I would encourage the authors to address in a revised manuscript.

---

## Author Comment (AC2) · 15 Oct 2020

**Response to Dr. Bernard Hallet (Reviewer #2) to manuscript TC-2020-82**

*Italic:* Referee comments
**Bold:** Authors comments

Referee:

*This is a fine paper, clearly presented and well illustrated, but only with skeletal captions that do not do justice to the figures...*

Authors:

**Incorporating also all the following comments, the figure captions have now been updated.**

Referee:

*One important improvement would be to add credibility to the ages reported by providing more explicit details about the impact on the calculated age of the initial content of Thorium 230 in the precipitate. One effective way of doing this is in table form much as that shown below from Fitzpatrick, J. J., Muhs, D. R., & Jull, A. J. T. (1990). (Saline minerals in the Lewis Cliff ice tongue, Buckley Island quadrangle, Antarctica. Contributions to Antarctic Research I, 50, 57-69). In particular, for 230/232 values of 4, for example, the age could be as much as 40% younger than the age calculated that does not assume there is any 230 initially. Also, the text should reflect as accurately as possible the corresponding large uncertainties.*

Authors:

**We added extra data to the text, stating that we assumed an initial 230Th/232Th ratio of 0.825 ± 50 % (the bulk-Earth value, which is the most commonly used for initial/detrital 230Th corrections). In addition, the reader is referred to the Supplementary table showing the detailed data and including the detailed analysis procedures, which are furthermore referenced. For example, the 230Th/238U and 234U/238U activity ratios of the samples were calculated using the decay constants given in (Cheng et al., 2000). The non-radiogenic 230Th was corrected using an assumed bulk-Earth atomic 230Th/232Th ratio of 4.4±2.2×10-6. U-Th ages were calculated using the Isoplot/Ex 3.75 Program (Ludwig, 2012).**

Referee:
*The authors may also wish to consider leveraging the limelight of Ötzi, the Iceage Man, and its climate implications, as referenced by Solomina et al (2015) in their supplementary material.*

Authors:

**A helpful suggestion, which we included now in the text with appropriate referencing.**

Referee:

*29. ...significant volume fraction of what?*

Authors:

**It is a significant ice volume fraction. We corrected this in the text.**

Referee:

*41. Also reported from the southern tip of S. America (Tierra del Fuego, Personal communication, Rabassa), New Guinea (Peterson and Moresby, 1979), and from sites where they formed under LGM ice.*

Authors:

**We added New Guinea to the text, but, for now, left S. America out as it can only be cited as 'personal communication'. Nevertheless, we consider the reviewer's suggestion of S. America as very important, because it is assumed that the lack of available 'published material' is a likely cause that subglacial carbonates in S. America are 'missing'.**

Referee:

*56-57. The units, $kg/m^2/yr$, seem unusual. Why not report ice thinning rate in m/yr, or the rate of increase of exposed bedrock, $m^2/yr$? This rate must be averaged over a certain area, but what is it? This reference, Gabrovec et al., 2014, does not help; it is incomplete and insufficient.*

Authors:

**We changed the units and used and cited the newly published data about the Triglav Glacier retreat (…around 0.6 m/yr (1952-2016) (Triglav-Čekada and Zorn, 2020)).**

Referee:

*Fig. 1 caption should be more informative, explaining to unfamiliar readers*
- *what is what (bedrock vs. precipitate)?*
- *the orientation of surface imaged relative to horizontal and to the former sliding direction*
- *the morphology of the precipitates*

Authors:
**We updated the figure caption.**

Referee:

*Fig. 2. What are is the pink areas? Replace these terms in legend; in English they are incorrect or awkward.*

> *From:*
> **Relief types**
> *Erosional topography*
> *Depositional topography*
> *Periglacial topography*
>
> **Relief Shapes**
> *Main ridge*
>
> *To:*

***Terrain types***
*Erosional surfaces*
*Depositional surfaces*
*Periglacial terrain*

***Relief Elements***
*Topographic divides*

Authors:

**The terms have been corrected accordingly – note that the whole Figure 2 has been extensively edited, and eventually joined with the Figure S2, which makes the message of the figure itself more condensed and clearer. We omitted the topography types as their boundaries cannot be strictly defined (in places they are coinciding), and rather made the glacier extents throughout the history clearer.**

[Figure]

Referee:

*60. Replace "…the recently exposed subglacial carbonate deposits due to glacier retreat." by "…the subglacial carbonate deposits recently exposed by glacier retreat."*

Authors:

**Corrected.**

Referee:

*78. (…targeting carbonate) cement ought to be replaced by the more appropriate word, precipitate*

Authors:

**Corrected.**

Referee:

*103. Replace "They are fluted and furrowed crust-like deposits characterized by brownish, greyish or yellowish colour." by "The fluted and furrowed crust-like deposits are brownish, greyish or yellowish in colour."*

Authors:

**Corrected.**

Referee:

*115-6. "Depending on the angle of the lee side of bedrock protuberances, columnar calcite crystals grow either perpendicularly to the host rock (Fig. 5a) or with a lower angle, generally oriented downslope…" Replace "angle" by "inclination" or "slope". It would be good to explain how the crystal orientation varies with bedrock surface inclination less ambiguously. For example, do vertical crystals grow perpendicular to the rock when bedrock surface is near vertical or near horizontal? Is there a relation between the crystal orientation and the former glacier sliding direction?*

Authors:

**To avoid ambiguity, we have changed the cited sentence to the following "Calcite crystals grow perpendicularly to the substrate on the steeper, nearly vertical areas of the bedrock (Fig. 4a) while in the less steep, nearly horizontal areas, crystals grow inclined, oriented downslope, presumably in the sliding direction of the forming glacier (Supp. Fig. S6)."**

Referee:

*124. Replace is by are.*

Authors:

**Corrected.**

Referee:

*127. For the isotopic ratios, the ranges should be included in the text, as well as averages. The reader should not have to look up the supplement for this basic information.*

Authors:

**Corrected.**

Referee:

*129-134.  Briefly explain what ages you expected.  Weere the "two U-Th ages of stratigraphically younger cement" obtained from the same sample?   If any of the thin sections are from this sample, you should mention it in the text, and help understand the stratigraphic setting of these younger deposits. Why would the former glacier be thick, and what do you mean by thick?*

Authors:

**The expected ages were of Little Ice Age, as it was assumed that glacier was completely melted during the Holocene Climactic Optimum – we added this in the text. We also referred readers to Figure 3 for more details where sample were obtained from (drilling locations), which also explains 'stratigraphically younger/older' situations. In the main text, we changed "of stratigraphically younger cement of the thickest …" to "corresponding to samples drilled in more surficial calcite layers of the thickest". By 'thick -glacier-' we mean 'of sufficient thickness to cause regelation', which is now clarified in the text.**

Referee:

*147. I'd replace low supersaturated solutions by slightly supersaturated solutions*

Authors:

**Corrected.**

Referee:

*148-9.  Replace "high Mg/Ca ratio in the water partially could be the trigger for the precipitation of aragonite" by "high Mg/Ca ratios in the water partially be responsible for the precipitation of aragonite".*

Authors:

**Here we have rewritten the sentence to make it clearer. The new sentence is: "In some freshwater systems like spring deposits (Jones, 2017), and specially in speleothems, Mg/Ca ratios seem to be the main factor controlling aragonite vs calcite precipitation (Frisia et al. 2002; Wassenburg et al. 2012; Rossi and Lozano, 2016)".**

Referee:

*150-1.  Recast sentence to avoid circular logic.*

Authors:

**Corrected.**

Referee:

*152. I suggest replacing "pose a challenge to determine" by "raise the difficult question of"*

Authors:

**Corrected.**

Referee:

*159-160. What is it about the moment that matters in the following "... factors such as the percentage of initial aragonite and the moment of the aragonite to calcite transformation...the possible additional redistribution of Th .... or the degree of opening of the system"? How about this rewording: "... factors such as the initial relative amount of aragonite and the timing of its transformation to calcite ...the possible additional redistribution of Th .... and the extent of chemical exchange with widespread subglacial meltwater"?*

Authors:

**Corrected.**

Referee:

*162-4. Suggested edit from:*
*"Based on the U concentration in samples within this study (in ppm; Supp. Table S2), it is notable that the youngest sample (2ka; T.03_b1) has 1.77 ppm of U concentration, whilst two of the old samples (LGM and YD; T.01_a1 and T.03_a1) have around 0.41 and 0.46 ppm of U concentration, respectively"*
*To*
*"It is notable that the U concentration  (in ppm; Supp. Table S2) in the youngest sample (2ka; T.03_b1) within this study is 1.77 ppm, whereas it is around 0.41 and 0.46 ppm in two of the old samples,  T.01_a1 and T.03_a1, respectively LGM and YD."*

Authors:

**Corrected.**

Referee:

*165. Replace contrary by to the contrary*

Authors:

**Corrected.**

Referee:

*166 & 168. Replace In case of the first possibility...by Assuming the first possibility...The same goes for line 168.*

Authors:

**Corrected.**

Referee:

Figure 4:
An informative caption is needed for this important figure. The labels are ambiguous. I assume, but am uncertain, that upper and lower "sides" refers to the surface (facing open air before being collected) and underside of the carbonate samples. In any case, how is depth measured? Is it relative to the surface or to the underside? For T.03, does the "lower side" include limestone bedrock as well as subglacial precipitate? It would be helpful for the reader to indicate clearly in words or graphics how the ages vary with stratigraphy. Perhaps, this could be dome easily by providing the ages that correspond to the depths written on figure. Is this the only sample for which several dates have been obtained?

Authors:

**Upper and lower sides refer to the surface (facing open air before being collected) and underside of the carbonate samples, respectively (the most appropriate side was chosen for drilling and consequently for visual presentation within this figure). The depths were measured relative to the starting drilling point of each sample as they are shown in this figure (lower side perspective for sample T.03 and upper side perspective for all the other samples); only the T.03 sample had precipitates thick enough that allowed to obtain several dates in three different depths. We also made the figure clearer in terms of differentiating what is subglacial carbonate and what is limestone bedrock.**

Referee:

*192. $\delta 18O$ differences of a few per mill in the carbonate precipitate can also arise due to variations in subglacial hydrology shifting from closed to open geochemical systems (see Hanshaw & Hallet)*

Authors:

**Corrected.**

Referee:

*197. Replace "lighter in deuterium compared to the Triglav…" by "lighter in deuterium than those from Triglav…"*

Authors:

**Corrected/**

Referee:

*199. The last clause (constraining the implications that the Triglav Glacier was constant during the Holocene.) does not follow logically from the preceding text. Clarify or delete it.*

Authors:

**Corrected.**

Referee:

*203. I am unsure of your intent with the leading sentence of this section. If it is consistent with the heading, I would suggest this revision: The LGM and YD ages are the first physical evidence that Triglav Glacier persisted through the Holocene to the present day (Solomina et al., 2015). If your intent is different, describe it clearly.*

Authors:

**We made the leading sentence of this section clearer.**

Referee:

*210. I suggest replacing "not being documented in the literature" by "not having been reported in the literature"*

Authors:

**Corrected.**

Referee:

*214. This paragraph ought to be updated in view of more recent work reporting even slower denudation (e.g. Steinemann et al., 2020), but still supports the contention that the 5mm crust would have weathered off during the HCO if exposed to the elements.*

Authors:
**The recent work of Steinemann et al. (2020) has been incorporated into the text.**

Referee:

*221. Delete the last 3 words because once exposed the subglacial carbonate deposits they cannot be glacially abraded. Moreover, I would not expect them to be abraded even under the glacier because they form in lee positions where I would expect abrasion to vanish as abrading rock fragments diverge from the bed at sites of subglacial precipitation due to regelation ice growth.*

Authors:

**Corrected.**

Referee:

*228. Delete "assessment of"*

Authors:

**Corrected.**

Referee:

*240. Replace "quick rate of 21st" by "rapid 21$^{st}$"*

Authors:

**Corrected.**

Referee:

*243. Replace "show a lower sensitivity to climate fluctuations" by "are less sensitive to climate fluctuations"*

Authors:

**Corrected.**

Referee:

*245. This remark about bright limestone substrate reminds me of the photographs, which begs the question: what is responsible for the color difference (greys vs. beige & brown on the more recently exposed bedrock surfaces)?*

Authors:

**To our knowledge, this has not been studied on that particular location. In general terms, the discolouration of the limestone is usually due to microbial (and lichen) activity (e.g. Dias et al., 2018). The more recently exposed bedrock surfaces have, assumingly, been affected less (for the shorter time) by microbials than the long-exposed bedrock in further surroundings. We inserted this remark in the Figure 1 caption.**

Referee:

*250. What do you mean here? The preliminary data shows a high possibility that subglacial carbonate deposits may endue unprecedented retreat… Might you mean the following? The preliminary data suggest that subglacial carbonate deposits can archive valuable datable records of glacial retreat, including hints that the current and ongoing retreat is unprecedented.*

Authors:

**Yes. We paraphrased the sentence to make the statement clearer.**

Referee:

*255. Replace observe by determine.*

Authors:

**We replaced 'observe' by 'detect'.**

Referee:

*260. Replace considerably fast by relatively high.  This section should also leverage, and be updated by, recent work (Steinemann et al., 2020).*

Authors:

**Corrected. We also referred the reader to the Section discussion limestone denudation and glacial erosion on limestone substrate, which includes recent work by Steinemann et al., 2020.**

Referee:

*264-5.  Replace "particulates on the present remnants of ice (and possible ice cores, if a glacier has not disappeared completely)" by "material on the small remaining ice masses and, if possible, ice cores)*

Authors:

**Corrected.**

Referee:

*274.  The conclusion would be stronger and clearer without the first sentence. I would replace it as follows: Subglacial carbonate deposits recently exposed by the retreating Triglav Glacier contain the first direct evidence of the existence and extent of Triglav Glacier since the Last Glacial Maximum and Younger Dryas. The deleted sentences should be incorporated in the previous section and clearly explained:"U-Th ages of subglacial carbonate with the combination of aragonite and calcite are regarded as maximum ages as aragonite-to-calcite transformation, evident in fabrics, might have occurred in calcite crystals that could have been falsely considered as primary.*

Authors:

**We re-arranged the conclusion, especially the first sentence. The previous message of the sentence is a part of Section 4.**

Referee:

*Missing refs*
*Hanshaw, B. B. and B. Hallet. 1978. Oxygen isotope composition of subglacially precipitated calcite: possible paleoclimatic implications. Science, 200,1267-1270.*
*Peterson, J. A., and Moresby J.F. 1979 Subglacial travertine and associated deposits in the Carstensz area, Irian Jaya, Republic of InCorrectedsia. Zeitschrift fur Gletscherkunde und Glazialgeologie. 15(1), 23-29*
*Steinemann, O., Ivy-Ochs, S., Grazioli, S., Luetscher, M., Fischer, U. H., Vockenhuber, C., & Synal, H. A. 2020. Quantifying glacial erosion on a limestone bed and the relevance for landscape development in the Alps. Earth Surface Processes and Landforms, 45(6), 1401-1417.*

Authors:

**All these references are now included and cited in appropriate places within the text.**

Referee:

*Figure S2:*
*The caption states "The recently exposed surface with shafts and subglacial carbonate deposits" but does not address the distinct colors, ranging from greys to rusty brown. Do the color boundaries correspond to, or parallel outlines of glacial extent in Figure S4? The figure should also show where the subglacial deposits or shafts are. In fact, what do you mean by shafts?*

Authors:

**Figure S2 has been deleted as very similar photograph with more detail is shown in Figure 1. Colour boundaries generally correspond to the glacial extent in the 19th century, but note that deposition of graviclastic material (which is of brighter colour as well) took place in some areas, which correspond to the glacial extent coincidentally. However, in general it matches to the oldest extent documented, and we included the observation to the caption.**
**By shafts we mean vertical caves, which we added now in the text where the word 'shaft' first appear.**

Referee:

*Figure S8: XRD graphs. A short caption is needed to explain what these samples are, how they differ from one another, and which are bedrock (if any).*

Authors:

**We have improved the caption: "XRD diagrams of all the studied samples (for precise location, see Figure 3 in the main manuscript). Samples TRG-01 C and TRG-03 D correspond to the bed rock. Green lines mark the reflections corresponding to calcite and red lines, those corresponding to aragonite.".**

Referee:

*Comments and questions in italics. Figure S9: a) Short columnar calcite crystals alternating with brown micritic bands constitute the first phase of calcite precipitation on the bedrock. Plane polarised light (PPL); b) columnar calcite crystals predominantly oriented towards the right (downslope) .[what is the orientation of the thin section relative to the former sliding direction? Same question for Fig S10]. The growth of the crystal on the center crosscut the direction of growth of previous crystals.[what is the center crosscut?] Crossed polarised light (XPL)*

Authors:

**The figure captions have been modified according to the suggestions. The sentence referring to how some crystals crosscut the previous has been deleted and we have added information on the orientation of thin sections. "Thin sections orientation is parallel to the former glacier sliding direction.".**

Referee:

*Figure S11- caption needs a brief explanation of what the figure shows.  What are the various curves?*

Authors:

**Figure S11 represented a location photo and a graph of $^{14}$C results of dated moraine organic matter by Karsten Grunewald and his team in the laboratory in Erlangen. We initially included it in the paper as these results have not been published elsewhere. However, since the focus on the paper is on U-Th dates of subglacial carbonates and discussion about their preservation, and since discussion of dating techniques of moraines are not included, we feel that this figure rather confuses readers. We deleted this figure now.**